

# CO$_2$ and CH$_4$ fluxes are decoupled from organic carbon loss in drying reservoir sediments

Tricia Light[1,2], Núria Catalán[1], Santiago Giralt[3], and Rafael Marcé[1]

[1]Catalan Institute for Water Research (ICRA), Emili Grahit 101, 17003, Girona, Spain
[2]Scripps Institution of Oceanography, University of California San Diego, 8622 Kennel Way, La Jolla, CA 92037, California, USA
[3]Institute of Earth Sciences Jaume Almera (ICTJA-CSIC), Lluís Solé Sabaris s/n, 08028, Barcelona, Spain

**Correspondence:** Tricia Light (Tlight@ucsd.edu)

**Abstract.** Reservoirs are a prominent feature of the current global hydrological landscape, and their sediments are the site of extensive organic carbon burial. Meanwhile, reservoirs frequently go dry due to drought and/or water management decisions. Nonetheless, the fate of organic carbon buried in reservoir sediments upon drying is largely unknown. Here, we conducted a 45-day-long laboratory incubation of sediment cores collected from a western Mediterranean reservoir to investigate carbon

dynamics in drying sediment. Drying sediment cores emitted more CO$_2$ over the course of the incubation than sediment cores incubated with overlaying water (206.7 $\pm$ 47.9 vs. 69.2 $\pm$ 18.1 mmol CO$_2$ m$^{-2}$ day$^{-1}$, mean $\pm$ SE). Organic carbon content at the end of the incubation was lower in drying cores, which suggests that this higher CO$_2$ efflux was due to organic carbon mineralization. However, the apparent rate of organic C reduction in the drying sediments (568.6 $\pm$ 247.2 mmol C m$^{-2}$ day$^{-1}$, mean $\pm$ SE) was higher than C emission. Meanwhile, sediment cores collected from a reservoir area that had already been

exposed for 2+ years displayed net CO$_2$ influx from the atmosphere to the sediment (-136.0 $\pm$ 27.5 mmol CO$_2$ m$^{-2}$ day$^{-1}$, mean $\pm$ SE) during the incubation period. Sediment mineralogy suggests that this CO$_2$ influx was caused by a relative increase in calcium carbonate chemical weathering. Thus, we found that while organic carbon decomposition in newly dry reservoir sediment causes measurable organic carbon loss and carbon gas emissions to the atmosphere, other processes can offset these emissions on short time frames and compromise the use of carbon emissions as a proxy for organic carbon mineralization in

drying sediments.

## 1 Introduction

Reservoirs are a dominant feature of the current global hydrological landscape. There are already an estimated 16.7 million reservoirs over 0.01 ha in size worldwide, and new reservoir construction is accelerating (Lehner et al., 2011; Pekel et al.,

2016; Zarfl et al., 2015). In part due to their prevalence, reservoirs play a significant role in the inland water carbon cycle (Li et al., 2015; Deemer et al., 2016; Tranvik et al., 2009; Cole et al., 2007; Clow et al., 2015). Recent estimates suggest





that approximately 0.06 Pg of organic carbon is buried in reservoir sediments every year (Mendonça et al., 2017). Organic carbon burial in reservoirs can be orders of magnitude greater than that in terrestrial forest soils and ocean sediments (Dean and Gorham, 1998; Mendonça et al., 2017; Clow et al., 2015). Eutrophic reservoirs in particular exhibit high organic carbon burial (Downing et al., 2008).

Meanwhile, water bodies such as reservoirs frequently go dry (Donchyts et al., 2016). Thousands of reservoirs partially or completely dry every year due to drought and/or water management decisions (Ragab and Prudhomme, 2002). Moreover, drying will likely become even more widespread as global climate change causes drought events to become more frequent and extreme (Herring et al., 2015). Nonetheless, the fate of organic carbon buried in reservoir sediments that are exposed to the atmosphere in drawdown areas during reservoir drying is largely unknown.

Increased exposure to oxygen and the other biogeochemical changes that occur as sediment dries may cause organic carbon buried in reservoir sediments to decompose due to the higher energy yield of oxygen reduction relative to that of other electron acceptors. This decomposition may in turn release carbon gases to the atmosphere (Marcé et al., 2019). Investigations performed in similar transient freshwater ecosystems such as temporary ponds and streams show significant carbon emissions upon drying (Catalán et al., 2014; Gómez-Gener et al., 2015, 2016; Obrador et al., 2018). Likewise, an investigation of reser-

voir sediments during an extreme drought event in South Korea documented pulses of carbon dioxide emissions upon drying large enough to counteract years of preceding carbon burial (Jin et al., 2016). Furthermore, an incubation experiment that dried sediment cores collected from a Brazilian reservoir showed substantial $CO_2$ and $CH_4$ emissions during drying and a significant effect of recurrent atmospheric exposure on sediment carbon fluxes (Kosten et al., 2018). Thus, while some evidence suggests the relevance of carbon emissions during sediment drying, the scarcity of available data precludes a full understanding of the

mechanisms behind these emissions, limiting our knowledge of the responses to drying across sediment types and climates.

Carbon emissions are thought to be due to organic carbon decomposition, but direct evidence linking emissions to decreases in sediment organic carbon is still lacking. Traditionally, carbon emissions have been related to changes in the carbon stock (Brüggemann et al., 2011; Subke et al., 2006). However, a number of other processes, both biological (e.g. chemolithoautotrophic) and purely geochemical, may also occur during reservoir sediment drying and influence carbon gas fluxes. For

instance, shifts in the equilibrium of calcium carbonate can produce either carbon dioxide effluxes (Fa et al., 2016) or influxes (Hamerlynck et al., 2013; Fa et al., 2016; Emmerich, 2003; Xie et al., 2009; Chen and Wang, 2014; Lapenis et al., 2008) in arid soils, as described by the reaction:

$$CaCO_3 + CO_2 + H_2O \leftrightarrow Ca^{2+} + 2HCO_3^- \tag{R1}$$

Either process may affect sediment carbon fluxes, particularly in watersheds showing high calcium carbonate content in soil

horizons and bedrock. Thus, the validity of sediment carbon emissions as a proxy for organic carbon mineralization in drying reservoir sediments must be explicitly tested.

In this study, we assessed carbon fluxes and changes in sediment carbon content and mineralogy over a 45-day-long laboratory incubation simulating drying in a western Mediterranean reservoir. We hypothesized that drying sediment cores would



emit substantial amounts of $CO_2$ and that these emissions would be accompanied by a concomitant change in the organic carbon content of the sediments. We also expected our study site's calcite-rich sediment to result in minor deviations in the mass balance between carbon emissions and organic carbon loss due to changes in the sediment carbonate equilibrium.

## 2 Methods

### 2.1 Study site and sample collection

Sediment cores were collected from Siurana Reservoir in Tarragona, Spain. The reservoir catchment is mostly forested, and its geology is dominated by Mesozoic carbonates. Sampling was conducted in January 2018, at which time reservoir volume was 12.6% of capacity (1.51 hm$^3$ out of 12 hm$^3$). Reservoir volume had been consistently below 50% of capacity since Fall 2016 due to a combination of drought and water diversion. Fifteen cores were collected from two sites within the reservoir: an exposed location that had already undergone drying for at least two years (six cores) and a submerged location below approximately 0.5 m of water (nine cores). Sediment cores were collected using PVC liners with an inner diameter of 6 cm and length of 60 cm. Cores at dry locations were pushed or hammered manually into the sediment as far as possible before extraction and capping and for submerged locations, by using a UWITEC gravity corer. All submerged cores were collected along with at least 20 cm of overlaying reservoir water, and 10 additional liters of water were also collected from the submerged reservoir site. In situ $CO_2$ emissions at both locations were estimated measuring $CO_2$ build up in a soil respiration chamber or a floating chamber connected to a PP Systems EGM-4 Environmental Gas Monitor (Supplement Table S1). Samples were kept cold and dark and transported back to the lab. Water was filtered using pre-rinsed 0.1 $\mu$m PVDF filters (Millipore, Massachusetts, U.S.A.) and stored in a 4°C cold room until further use.

### 2.1.1 Incubation set-up

Cores were named according to their experimental treatment and initial collection location (i.e. "Initial: Control Wet", "Initial: Control Dry", "Incubation: Wet", "Incubation: Dry", and "Incubation: Wet-Drying" (Fig. 1). For the initial analysis experimental treatment, three cores from each location were randomly selected for analysis of initial organic matter content conditions (i.e. "Initial: Control Wet" and "Initial: Control Dry". The remaining six cores from the "Wet" location were randomly assigned to either "Incubation: Wet" or "Incubation: Wet-Drying" incubation treatments. The remaining three cores from the "Dry" location were assigned to the "Incubation: Dry" incubation treatment (Fig. 1). To set up the incubation, overlaying reservoir water was gently drained from both "Incubation: Wet" and "Incubation: Drying" cores. "Incubation: Wet" cores were immediately re-submerged under 10 cm of filtered reservoir water, while "Incubation: Drying" cores were incubated directly after draining. "Incubation: Dry" cores were incubated as collected (i.e. dry, with no overlaying water). All incubation cores were gently raised from below to have 10 cm of headspace over the sediment or water surface, depending on the treatment. All cores were placed uncapped and upright in a dark incubation chamber at 25°C (Figure S1). Desiccant beads were placed at the bottom of the chamber, and pressurized air was constantly circulated through the chamber throughout the 45-day-long incubation to ensure





01.pdf

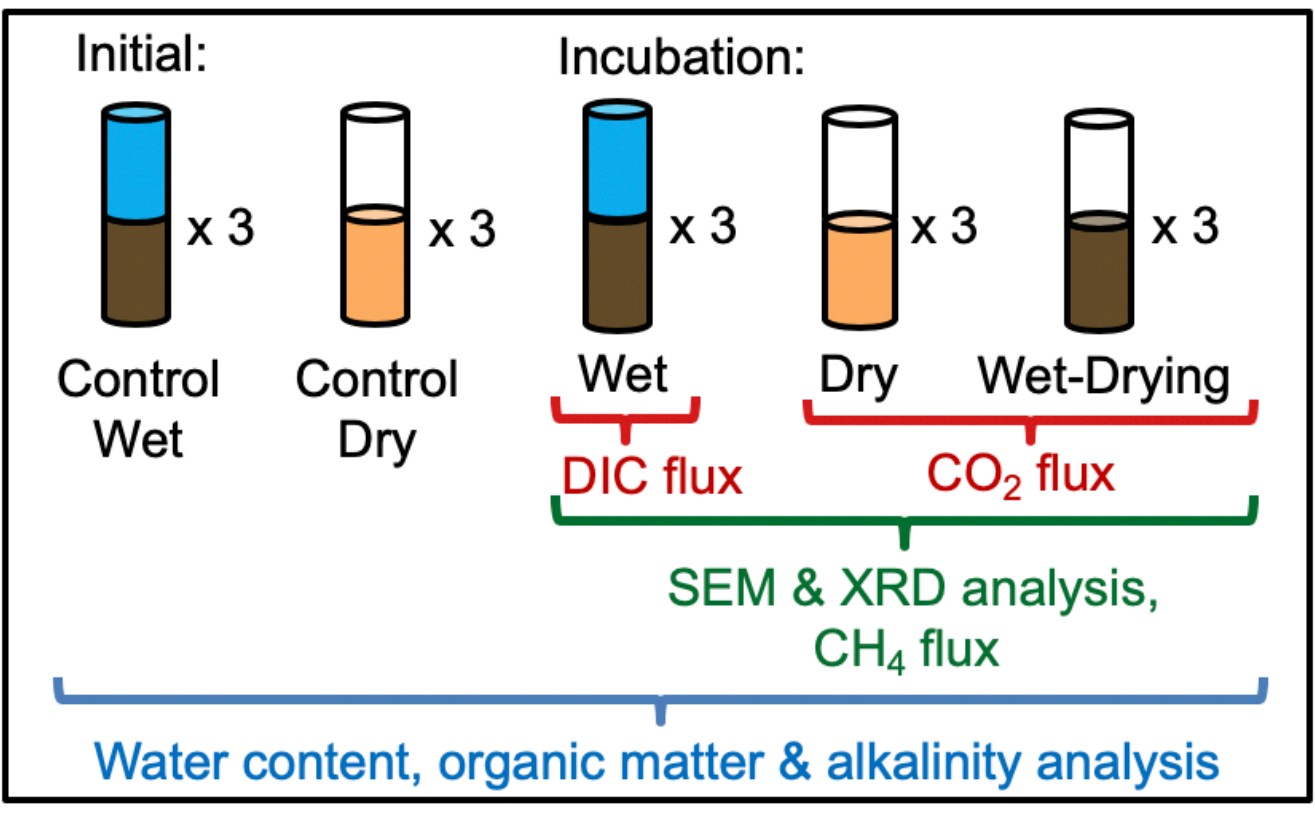

**Figure 1.** Methodological scheme for the collection, incubation, and analysis of sediment cores. Treatments with dark brown sediment represent cores collected from the "Wet" site, while treatments with light brown color represent cores collected from the "Dry" site. SEM: scanning electron microscopy; XRD: X-Ray diffraction.

a dry environment conducive to core drying, and to avoid $CO_2$ build-up inside the incubation chamber. Filtered water was periodically added to the "Incubation: Wet" cores throughout the incubation period to replace water lost through evaporation and maintain a constant water level.



### 2.1.2 Gas flux measurements

Core mass and fluxes of greenhouse gases were measured on the first day of the incubation and periodically thereafter. For the analysis of $CO_2$ flux to the overlying air, "Incubation: Dry" and "Incubation: Drying" cores were temporarily covered with air-tight, custom made caps. These caps created closed chambers with a surface area of 28.3 cm$^2$ and volume of 283 cm$^3$. The

chambers were connected to an environmental gas monitor (EGM-4, PP Systems, Massachusetts, U.S.A.) to directly measure $CO_2$ flux. $CO_2$ concentration within the chamber was analyzed every 4.8 seconds with an accuracy of 1%. Each chamber analysis lasted at least 300 seconds or until a 10 $\mu$atm change in $CO_2$ was recorded, whichever came first. $CO_2$ flux was measured approximately every other day.

    "Incubation: Dry" and "Incubation: Drying" cores were capped as described above for 30 minutes for $CH_4$ flux analysis

on days 0, 1, 5, 19, and 45 of the incubation. 10 ml air samples were collected via syringe through a septa at the beginning and end of chamber deployment and transferred to pre-evacuated glass vials (Exetainers 339 W, Labco Lim. Lampeter, UK). Samples were analyzed for $CH_4$ within 3 weeks of sampling using a gas chromatograph (7820A with a 77697 headspace sampler, Agilent, CA, U.S.A.). The gas chromatograph was calibrated at the beginning and end of each analysis session using a standard curve spanning 1-25 ppm $CH_4$ in a $N_2$ gas mixture, with a precision of 0.6 ppm $CH_4$.

For "Incubation: Wet" cores, $CO_2$ and $CH_4$ fluxes were also measured on incubation days 0, 1, 5, 19, and 45. The overlaying water of all cores was covered with an airtight plastic seal for 20 minutes. Water was collected via syringe immediately before and after the 20 minutes and analyzed for DIC as a proxy for sediment $CO_2$ flux. DIC was analyzed via catalytic oxidation using a TOC-VCSH analyzer (Shimadzu, Japan). Additional water samples were allowed to equilibrate with headspace in enclosed syringes (10 ml water to 10 ml air). The $CH_4$ concentration of the equilibrated air was analyzed as described above

for the other treatments.

    Drying over the course of the incubation was periodically measured by placing each "Incubation: Dry" and "Incubation: Drying" core on a balance to track change in core mass.

### 2.1.3 Sediment analyses

All cores were drained, sectioned and analyzed for water and organic matter content, either upon arrival to the lab for "Initial"

or following the incubation for "Incubation" cores. Cores were sliced into 12 sections (0-1, 1-2, 2-3, 3-4, 4-5, 5-7, 7-9, 9-11, 11-13, and 13-15 cm in depth). Water content was determined by drying an aliquot of each section in a 70°C oven until a constant mass was reached (Fig. S2). This dry sediment was then combusted at 450°C for 4 hours and re-weighed to determine the percentage of weight lost on ignition (LOI), a proxy for sedimentary organic carbon content. We assumed that organic carbon content was equivalent to LOI/2 (Dean and Gorham, 1998). All remaining sediment was frozen at -20°C. Sediment

from near the surface (1-2 cm in depth) for all cores was later defrosted and analyzed for pH and alkalinity using a Metrohm 848 Titrino Plus Titrator. Sediment was suspended in a 2:1 deionized water:sediment solution and filtered, and the pH of the resultant filtrate was analyzed.





Frozen incubation treatment sediment was freeze-dried for additional mineralogical analyses. Freeze-dried sediment was ground, and its mineralogical composition was determined using a Siemens D500 automatic X-Ray diffractometer (working conditions: Cu K-alpha, 40 kV and 30 mA). The identification of mineralogical species was carried out using EVA software attached to the diffractometer and their quantification was done using the standard procedure (Chung, 1974). The uncertainties associated to the quantification method are 5% wt. Albite, calcite, clinochlorite, dolomite, gypsum, kaolinite, microcline, muscovite, and quartz were identified and quantified. To qualitatively assess differences between treatments, freeze-dried sediment from 0-1 cm (i.e. shallowest) and 13-15 cm (i.e. deepest) was coated with gold and imaged using a scanning electron microscope JEOL J-6510 equipped with an EDS detector at the Scientific and Technological Centers of the University of Barcelona.

The influence of biological activity in carbon dioxide flux was assessed using defrosted sediment from 1-2 cm in depth from a randomly selected "Incubation: Dry" core. This sediment sample was split into 3 replicate sections of 5 ml each, each of which was placed in a separate 100 ml glass beaker and covered with an airtight seal. Baseline $CO_2$ flux from each section was analyzed as described above for sediment cores. Samples were then sterilized by exposure to UV light under a laminar flow hood (AH-100, Telstar, Catalonia, Spain) for 45 minutes followed by microwaving at 700 Hz for 90 seconds in 30 second increments, each separated by 1 minute of shaking. Sediment was then allowed to return to room temperature, re-sealed, and analyzed again for $CO_2$ flux.

## 2.2 Data analysis

All statistical analyses were conducted in R (R Core Team, 2018). To test differences in $CO_2$ flux, $CH_4$ flux, core mass, and water content between treatments, one-way analyses of variance (ANOVA) were conducted on mixed effect models with treatment considered as a fixed effect and replicate core within treatment as a random effect using the lmer function of the package nlme (Pinheiro et al., 2018). Depth was considered an independent variable for water mixed effect models, and time was considered an independent variable for $CO_2$ and $CH_4$ flux and core mass mixed effect models. To analyze organic carbon content differences between treatments, we first identified different sediment core layers as defined by a clustering analysis of the organic carbon content profiles from all cores. Clustering analysis was performed using the chclust function from the R package rioja and constraining the result by sample depth (Juggins, 2017). This analysis was performed to identify the depth of the surface layer affected by organic carbon changes during the incubation, for we expected organic carbon changes to be unlikely beyond a surface layer of unknown depth a priori. After identifying the depth of the surface layer most affected by organic carbon changes, we assessed differences in surface layer organic carbon content between treatments using ANOVA. Post-hoc Tukey tests were conducted to identify differences between treatments using the lsmeans package (Lenth, 2016).

Net $CO_2$ flux during the incubation was determined by performing trapezoidal integration under the curve of observed gas flux data points along time, using the trapz function in the package pracma (Borchers, 2018). To determine mineralogical trends among treatments, we conducted a principal component analysis (PCA) on the correlation matrix of the $arcsine\sqrt{(x)}$ transformed percent abundance data for each mineral using the prcomp function from R core. Pearson correlation tests were conducted between $CO_2$ flux and core organic carbon content, water content, and change in core mass, and between the first





two principal axes of the mineralogy PCA and organic carbon and water content using R core. Pearson correlation tests were also conducted between averages of the first two principal axes per replicate core and $CO_2$ flux and change in core mass. Extreme outliers were removed from all data sets following examination of box plots, Cook's influential outlier tests, and Cleveland boxplots (Zuur et al., 2010). All plots were created using ggplot from the tidyverse package (Wickham and Team, 2017).

## 3  Results

### 3.1  Incubation carbon gas fluxes

Gaseous carbon fluxes differed between each incubation treatment (F = 68.3, p < 0.001 for pairwise post hoc tests) (Table 1 and Fig. 2). "Incubation: Wet" incubation cores generally displayed positive $CO_2$ fluxes (i.e. out of the sediment) that declined in magnitude over time (Fig. 2). Meanwhile, "Incubation: Dry" cores displayed positive fluxes for the first week but then consistently negative (i.e. into the sediment) fluxes for the remainder of the incubation (Fig. 2). Post-incubation analysis showed that this $CO_2$ influx to the sediment persisted even after sediment sterilization. "Incubation: Wet-Drying" cores initially displayed positive $CO_2$ fluxes, but by the end of the incubation two out of three "Incubation: Drying" cores also displayed negative $CO_2$ fluxes (Fig. 2). Change in $CO_2$ flux over time significantly correlated with drying, as measured by decline in core mass (p = 0.004, $r^2$= 0.27). However, net $CO_2$ flux over the incubation period did not correlate with core-specific sediment organic carbon or water content.

Consistent nonzero mean $CH_4$ flux values were only observed for "Wet" and "Drying" treatments. Positive $CH_4$ fluxes were observed on Days 0, 1, and 6 and Day 1 for "Wet" and "Drying" treatments respectively (Figure S3).



**Table 1.** Summary of sediment core properties and incubation gas fluxes averaged across treatments types. All values are mean ± SE.

| | Initial: Control Wet | Initial: Control Dry | Treatment | | |
| --- | --- | --- | --- | --- | --- |
| | | | Incubation: Wet | Incubation: Dry | Incubation: Wet-Drying |
| Water Content (%) | 47.4 ± 0.2 | 23.8 ± 0.1 | 45.9 ± 0.2 | 7.8 ± 0.5 | 13.2 ± 0.3 |
| Alkalinity (/$\mu$M HCO$_3^-$) | 1.21 ± 0.12 | 0.42 ± 0.11 | 0.69 ± 0.06 | 0.42 ± 0.11 | 0.33 ± 0.01 |
| Average CO$_2$ Flux (mmol m$^{-2}$ d$^{-1}$) | - | - | 95.7 ± 14.4 | -134.4 ± 15.4 | 177.8 ± 21.6 |
| Average CH$_4$ Flux ($\mu$mol m$^{-2}$ d$^{-1}$) | - | - | 111.4 ± 47.8 | -1.5 ± 3.1 | 1.6 ± 6.3 |
| Net CO$_2$ + CH$_4$ Flux (mmol C) | - | - | 8.8 ± 2.3 | -17.3 ± 3.5 | 26.3 ± 6.1 |
| Organic Carbon Content (g/gdw %) | 3.52 ± 0.19 | 3.78 ± 0.19 | 2.68 ± 0.20 | 3.55 ± 0.20 | 2.70 ± 0.18 |





**Figure 2.** Sediment core $CO_2$ + $CH_4$ fluxes over the course of the 45-day-long incubation as determined by core headspace $CO_2$ and $CH_4$ measurements for "Incubation: Dry" and "Incubation: Drying" treatments, and overlaying water DIC and dissolved $CH_4$ measurements for "Incubation: Wet" cores. All replicates are shown in the graph. $CH_4$ fluxes composed less than 1% of carbon gas fluxes for all data points (see Fig. S3 for just $CH_4$ fluxes). Color lines are splines fitted to the data, and are included only for visual reference.




03.pdf

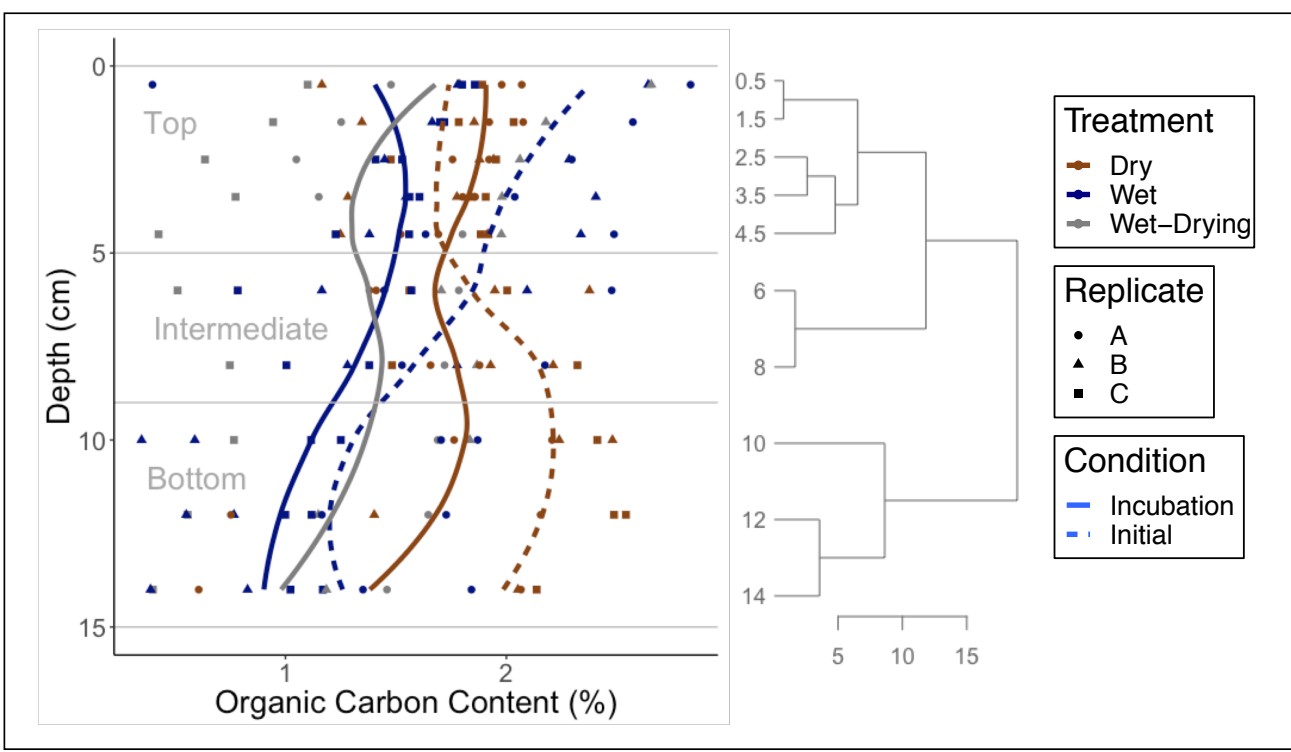

**Figure 3.** LOI sediment organic carbon content as a function of depth. Results of the clustering analysis are presented in gray, with top, intermediate, and bottom depth layers identified in the organic carbon content data. Data outliers were removed according to the criteria presented by Zuur et al. (2010). Colored lines are splines fitted to the data, and are included only for visual reference.

### 3.2 Sediment carbon loss

Clustering analysis for organic carbon content identified three main layers in the cores: a top layer between 0 and 5 cm, an intermediate layer between 5 and 9 cm, and a bottom layer between 9 and 15 cm (Fig. 3). Organic carbon differed between the five treatments (F = 8.52, p < 0.0001) and the three layers (F = 5.77, p = 0.004), with "Incubation: Drying" and "Incubation: Wet" cores displaying lower organic carbon content (g/g dw) than the other treatments (Table 1). In the top layer (0-5 cm), the only significant (p < 0.01) difference in post-hoc comparisons was that "Initial: Wet" was lower than "Incubation: Wet-Drying" (difference of 1.37% ± 0.34, p=0.0008). There was no significant difference in the intermediate layer (5-9 cm). In the bottom layer (9-15 cm), "Initial: Dry" was larger than "Incubation: Wet-Drying" (difference of 1.94% ± 0.44, p=0.0002), "Initial: Wet" (difference of 1.86% ± 0.44, p=0.0005), and "Incubation: Wet" (difference of 2.37% ± 0.44, p<0.0001) treatments. Similarly, sediment alkalinity was also highest in "Incubation: Wet" cores (F = 24.6, p < 0.001; Table 1).





### 3.3 Mineralogical sediment transformations

X-ray diffraction results suggested mineralogical transformations occurred during drying (Figs. 4, S4). A PCA run using percent abundances of the nine identified minerals revealed divergence between treatments (Figure 4). The first two axes of the PCA explained 74.51% of variance. The first axis of the PCA explained 45.6% of variance, with a positive loading of calcite and kaolinite and a negative loading of clinochlorite and quartz. The second axis accounted for 28.8% of variance, with a positive loading of quartz and negative loadings of dolomite, muscovite, and kaolinite. "Incubation: Wet-Drying" core samples were grouped by high quartz and clinochlorite, "Incubation: Dry" cores were grouped by high muscovite and dolomite content, and "Incubation: Wet" cores were grouped by high calcite. The first PCA axis scores correlated with organic carbon content ($p < 0.001$, $r^2 = 0.14$) and water content ($p < 0.001$, $r^2 = 0.29$), and the second PCA axis scores correlated with organic carbon content ($p = 0.002$, $r^2 = 0.13$). Average first and second PCA axis scores did not correlate with either $CO_2$ flux or change in core mass.



**Figure 4.** Principal component analysis (PCA) of mineralogy data for sediment samples from varying depths of "Incubation: Wet", "Incubation: Dry", and "Incubation: Wet-Drying" cores. Color identifies the treatment, while the size of the dots represent core sample depth.





SEM images of the sediment samples also showed differences between treatments. Calcium carbonate crystals in sediment collected from "Incubation: Wet" cores presented almost intact euhedral rhomboidal morphologies (Figs. 5a and b), while those from "Incubation: Dry" (Figs. 5d and e) and "Incubation: Wet-Drying" cores (Figs. 5g and h) showed significant corrosion at the crystal faces. The morphology of calcium carbonate crystals in "Incubation: Wet" cores were typically of planktonic origin

5  and likely originated from precipitation during diatom blooms (Fig. 5c. Meanwhile, most calcium carbonate in "Incubation: Dry" and "Incubation: Wet-Drying" cores was present as a thin coating covering all sediment surfaces (Figs. 5f and i). This calcium carbonate coating was likely linked to precipitation and chemical weathering cycles within the sediment in response to flooding and drying.





05.pdf



**Figure 5.** SEM imagery of sediment particles from "Incubation: Wet", "Incubation: Dry", and "Incubation: Wet-Drying" incubation treatments. Images a,c,d,e,g,h, and j are from sediment samples collected from 0-1 cm in depth, and images b and f are from sediment samples collected from 13-15 cm in depth.



## 4  Discussion

### 4.1  Carbon gas efflux in drying sediment

"Incubation: Wet-Drying" sediment cores consistently emitted $CO_2$ during the first month of the incubation (Fig. 2). However, the $CO_2$ efflux observed was short-lived: on average, $CO_2$ flux decreased at a rate of $14.2 \pm 3.7$ $\mu$mol m$^{-2}$ d$^{-1}$. Efflux was steadily decreasing for all "Incubation: Wet-Drying" cores by Day 14. $CO_2$ fluxes followed a consistent pattern between replicates up until Day 38, at which point two of the three replicates began displaying negative (i.e., into the sediment) $CO_2$ fluxes. This contrasts with $CO_2$ fluxes from "Incubation: Wet" treatment cores, which were consistently positive across all replicates throughout the incubation.

The initial positive $CO_2$ flux observed in the "Incubation: Wet-Drying" treatment ($170.6 \pm 52.9$ mmol m$^{-2}$ d$^{-1}$) was similar in magnitude to the effluxes observed in other drying freshwater sediments (reservoir: 137.1 mmol m$^{-2}$ d$^{-1}$ (Kosten et al., 2018); temporary pond: $121.3 \pm 138.1$ mmol m$^{-2}$ d$^{-1}$ (Catalán et al., 2014); streambed: $209 \pm 10$ mmol m$^{-2}$ d$^{-1}$ (Gómez-Gener et al., 2016)). However, this efflux was lower than that observed in other drying reservoir sediments ($900 \pm 150$ mmol m$^{-2}$ d$^{-1}$, Jin et al. 2016). This implies significant geographic variability in carbon fluxes during reservoir drying, as suggested by Marcé et al. (2019).

Similarly, the low methane fluxes observed in this study stress the relevance of local sediment properties in controlling carbon gas fluxes from sediment. Here, "Incubation: Wet-Drying" sediment cores only displayed nonzero $CH_4$ flux on Day 0 (Fig. S3). This positive $CH_4$ flux ($28.1 \pm 2.0$ $\mu$mol m$^{-2}$ d$^{-1}$) was much smaller than both the $CO_2$ efflux observed here and the $CH_4$ effluxes observed in other reservoir drying studies (Jin et al., 2016; Kosten et al., 2018). This suggests that site-specific sediment properties such as organic carbon content or grain size and porosity promoting oxic conditions during drying prevented significant methanogenesis from occurring.

### 4.2  Drying sediment carbon loss

Sediment organic carbon data suggests that the $CO_2$ efflux in "Incubation: Wet-Drying" sediment cores was driven by organic matter decomposition. Statistical analyses showed that "Incubation: Wet-Drying" cores displayed lower organic carbon content than "Wet" cores (Table 1, Fig. 3). If this small-scale experiment was representative of in-situ reservoir drying, the carbon loss via organic matter decomposition implied by this discrepancy has significant implications for the reservoir's carbon budget. The difference in organic carbon content between "Incubation: Wet" and "Incubation: Wet-Drying" treatments corresponds to an average organic carbon loss rate of $0.57 \pm 0.14$ mol m$^{-2}$ d$^{-1}$ over the course of the incubation and a net organic carbon loss of $3.07 \pm 0.76$ Mg ha$^{-1}$. This loss over just 45 days is comparable in magnitude to changes in soil organic carbon stock during the transition from tropical secondary forest to perenial crops (Don et al., 2011). Moreover, it is equivalent to reversing approximately 1 year of carbon burial at the average burial rate of 250 g C m$^{-2}$ yr$^{-1}$ reported by Mendonça et al. (2017) for inland waters. This significant carbon loss may undermine the notion of organic carbon burial in reservoirs as a long-term carbon sink, particularly in regions such as the western Mediterranean in which reservoirs are fairly dynamic ecosystems. Instead, decomposition during prolonged drying events may mineralize a sizeable fraction of the organic carbon buried in





sediment during the reservoir's lifetime. Thus, sediment carbon burial should not necessarily be considered a carbon sink in a reservoir's long-term carbon budget, specially in regions where drying events are expected to become more frequent in the future.

However, the large variability between replicate cores suggests significant spatial heterogeneity in sediment composition and highlights the need for spatial replication within and across reservoirs in future studies. The previously discussed evidence for organic carbon loss during drying implies that the initial carbon content would be lower in "Initial: Dry" cores than in "Initial: Wet" cores, but this was not the case (Table 1, Fig. 3). "Initial: Dry" sediment did not significantly differ from "Initial: Wet" sediment in organic carbon content. Considering the large variability among replicate cores collected from the same location (which would therefore be more accurately described as pseudo-replicates), greater spatial replication within the reservoir would likely be necessary to resolve differences in sediment carbon content between wet and dry sites. Therefore, although our dataset supports the presence of an enhanced mineralization process during drying, the potential implications at the whole water body scale cannot be fully resolved.

### 4.3 Decoupling of carbon gas efflux from sediment carbon loss

While the observed organic carbon loss from "Incubation: Wet-Drying" cores was consistent with the observed $CO_2$ fluxes in direction, approximately three times more organic carbon was lost than $CO_2$ was emitted. The difference in sediment organic carbon content in the surface layer (0-5 cm) of "Initial: Wet" and "Incubation: Wet-Drying" cores would correspond to a net efflux of $72.3 \pm 17.8$ mmol $CO_2$ (mean $\pm$ SE). However, observed net efflux was only $26.3 \pm 6.1$ mmol $CO_2$ (mean $\pm$ SE) per core. Thus, even considering the aforementioned variability in sediment organic carbon data, it appears that a significant portion of the organic carbon consumed via decomposition was not emitted as $CO_2$ but rather consumed by one or more other processes. This is also supported by the observed influx of $CO_2$ from the atmosphere into the sediment for "Incubation: Dry" treatment cores. Consistent $CO_2$ influxes in "Incubation: Dry" cores similar in magnitude to the $CO_2$ effluxes in "Incubation: Wet-Drying" cores were observed after one week of incubation across all replicates (Table 1, Fig. 2). Furthermore, by the end of the incubation two out of three replicate "Incubation: Wet-Drying" cores also displayed $CO_2$ influxes. These findings show the relevance of the $CO_2$ consumption pathway(s) active in these sediments. They also imply that the $CO_2$ effluxes observed in "Incubation: Wet-Drying" cores must be considered the net result of $CO_2$ production and consumption processes.

### 4.4 Sediment carbon consumption via calcium carbonate chemical weathering

Sediment mineralogy results suggest that the observed sediment carbon consumption was likely caused by an increase in calcium carbonate chemical weathering relative to precipitation (Reaction R1). "Incubation: Wet" sediment contained more calcium carbonate than either "Incubation: Dry" or "Incubation: Wet-Drying" cores (F = 10.7, p < 0.001; p < 0.01 for post-hoc tests; Figs. 4 and SXXX). This suggests a relative increase in calcium carbonate chemical weathering in dry conditions. Similarly, sediment gypsum ($CaSO_4 \cdot 2H_2O$) abundance data is consistent with calcium carbonate chemical weathering. Gypsum was only present in "Incubation: Dry" and "Incubation: Wet-Drying" cores (Figure SXXX). This indicates that it precipitated



during sediment drying, which is consistent with the elevated pore water $Ca^{2+}$ ions expected to accompany an increase in calcium carbonate dissolution.

SEM imagery provides further evidence of sediment calcium carbonate chemical weathering on the time-scale of the incubation. Calcium carbonate crystals in the "Incubation: Wet" treatment were euhedric, but crystals in the "Incubation: Dry" and

"Incubation: Drying" treatments were visibly corroded (Figure 5). Similarly, most carbonate in the "Incubation: Wet" treatment was present in the form of discrete crystals and visually biogenic in origin, while most carbonate in the "Dry" and "Drying" treatments appeared as a thin calcium carbonate coating covering all sediment surfaces. This suggests the existence of calcium carbonate precipitation and chemical weathering cycles, probably occurring as a response to sediment drying and flooding cycles. This supports the hypothesis that an increase in chemical weathering relative to precipitation may occur later in the

drying process due to the common-ion effect.

Few soil or sediment science investigations link sediment $CO_2$ influx to an increase in calcium carbonate chemical weathering relative to precipitation. Those that do generally link chemical weathering to factors that are not applicable in the context of this investigation, i.e. climate (Lapenis et al., 2008)), high sediment alkalinity (Lapenis et al., 2008; Emmerich, 2003; Xie et al., 2009; Wang et al., 2016; Ma et al., 2014), and diurnal cycling (Roland et al., 2013; Hamerlynck et al., 2013; Chen and

Wang, 2014; Fa et al., 2016). This raises the question of what conditions caused the calcium carbonate chemical weathering hypothesized to occur here. One possible explanation is a combination of 1) high carbonate sediments ($29.6 \pm 1.4$ % $CaCO_3$ mean $\pm$ SE for wet cores) and 2) high air flow and thus $CO_2$ availability due to sediment dryness. The role of dryness in promoting air flow would explain the lack of both chemical weathering in "Incubation: Wet" sediments and $CO_2$ influx to dry sediments during the first week of the incubation. Core collection was performed within 48 hours of a rain event, so even

exposed sediments were relatively humid at the beginning of the incubation. We posit that sediments may need to reach a certain dryness threshold to establish sufficient air flow and thus $CO_2$ availability for chemical weathering to occur. Under this scenario, sediment humidity is crucial to determining chemical weathering; sediments must be dry enough to establish sufficient air flow but humid enough for water to be available for the chemical weathering reaction to proceed.

Regardless of the precise mechanism(s) causing chemical weathering, high sediment calcium carbonate content and in-

termittently dry conditions are the most likely driving factors in this context. Thus, this process may regularly occur in this western Mediterranean reservoir as well as in similar systems around the world. Drying reservoir and lake sediments are understudied, so the calcium carbonate chemical weathering and precipitation observed here may be prevalent in a wide variety of contexts. Given the limited spatial replication and laboratory nature of this investigation, further work is needed to determine the relevance of this process under natural conditions.

**4.5 Implications for drying reservoir carbon dynamics**

The decoupling of organic carbon loss from $CO_2$ efflux and proposed consumption of inorganic carbon via calcium carbonate chemical weathering in reservoir sediments may have important implications for our understanding of sediment carbon dynamics. First, it indicates that the common strategy of equating carbon gas flux with organic carbon decomposition may be flawed. Thus, if calcium carbonate chemical weathering in sediments is geographically widespread, organic carbon mineraliza-



tion rates in dry sediments may be significantly underestimated by studies that only measure $CO_2$ efflux as a proxy for carbon mineralization. Although we cannot make conclusions regarding the prevalence of this decoupling in other reservoirs with our experiment, our findings constitute a warning that further research is necessary to understand the significance of this process to overall freshwater carbon cycling.

Further research on the fate of the alkalinity produced by the calcium carbonate chemical weathering process is also needed to determine its impact on the carbon budget of lakes and reservoirs that seasonally or permanently dry. While $CaCO_3$ chemical weathering decreases $CO_2$ efflux, it is unlikely to constitute a long-term carbon sink if the bicarbonate ions produced by dissolution eventually transfom to $CO_2$ and re-enter the atmosphere through equilibration (Wang et al., 2016). However, if the bicarbonate ions produced by $CaCO_3$ dissolution are sequestered in either sediment or groundwater, it is also possible that

the $CO_2$ influx observed in this study constitutes the basis of a previously unrecognized long-term carbon sink. The rate of carbon dioxide intake shown here is comparable to rates from a variety of Mediterranean and temperate forest soils (Baldocchi et al., 2018). Such a large sediment carbon sink would therefore carry considerable implications for our understanding of the freshwater carbon cycle, and this question also merits further research.

## 5    Conclusions

This investigation used a laboratory sediment core incubation to explore the effects of reservoir drying on sediment carbon dynamics. We directly linked organic carbon loss to carbon dioxide emissions in drying reservoir sediment for the first time, undermining the idea that organic carbon burial in active reservoir sediments represents a long-term carbon sink. However, we also found a decoupling between carbon loss and carbon gas fluxes and observed carbon dioxide influxes to most sediment cores analyzed. Mineralogical sediment composition suggests that these discrepancies were due to an increase in calcium

carbonate chemical weathering. Together, these findings show that while reservoir sediment drying can cause organic carbon decomposition and thus carbon gas efflux to the atmosphere, other sediment processes can potentially offset or even reverse these fluxes.

*Data availability.*   All raw experimental data can be found in the Supplement Tables S3-S10.

*Competing interests.*   The authors have no competing interests to declare.

*Acknowledgements.* TL was supported by Fulbright Research grant from Fulbright España. NC held a Beatriu de Pinós grant (BP2016-00215). This is a contribution to the project C-HydroChange (CGL2017-86788-C3-2-P), funded by the Spanish Ministry of Science, Innovation, and Universities.





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
