# Peer review of "$CO_2$ and $CH_4$ fluxes are decoupled from organic carbon loss in drying reservoir sediments"

_Biogeosciences, 2019_

## Referee Comment (RC1) · Anonymous Referee #1 · 4 Aug 2019

This paper reported results of incubation experiment for reservoir sediments. The authors found that CO2 emitted from organic matter degradation partly weathers carbonates in sediments, which apparently reduces CO2 outgassing into the atmosphere. This sounds like very interesting to me and the authors did a really good job. I was wondering if the authors could a little bit more expand their discussion to the global carbon cycling in a drying world. To what extent do the authors think should the current CO2 budget be revised if the carbonate weathering under the reservoir dryness is taken into account? Is this a trivial or negligible effect on the increasing temperature on this planet? Maybe one or two paragraph discussion would be beneficial for future researchers to follow this great work.

---

## Referee Comment (RC2) · Anonymous Referee #2 · 29 Aug 2019

'CO2 and CH4 fluxes are decoupled from organic carbon loss in drying reservoir sediments" by Light et al describes the contrasting roles of organic carbon metabolism and chemical weathering in reservoir sediments under contrasting hydrological conditions. This manuscript adds to a growing body of work that quantifies the joint effects of organic and inorganic carbon cycles on carbon emissions from freshwater ecosystems. Uniquely, this manuscript describes the contribution of calcium carbonate weathering to carbon emissions in a reservoir experiencing a long-term draw down.

General comments

Greater consistency in naming conventions would improve readability. For example, the names Incubation: drying and Incubation: wet-drying are inconsistently used throughout the paper. Also, using calcium carbonate instead of calcite (if there is a reason to

use both terms, an explanation would help!).

I appreciate that the authors' address the high variability among replicate cores and that they call for greater spatial sampling across the reservoir. To address the limitation of only sampling within one small region of the reservoir, I would like to see some more information about the location in the reservoir the cores were collected (inflow, transition zone, or outflow). In the Siurana Reservoir is there a transition from more inorganic, watershed derived sediments near the inflow and more organic sediments near the outflow? How does that relate to your findings?

Specific comments 1) Page 2, line 24. Can you expand on the environmental conditions in which the equilibrium reaction of calcium is important and how those conditions relate to the environmental conditions in reservoirs?

2) Page 3, line 10. How certain are you that the wet sites have been consistently wet for the last 2 years? How would intermittent drying affect your results?

3) Figure 1. Are the DIC-method and flux methods generally consistent for the Incubation: Wet treatments?

4) Page 6, line 10. What are your findings? Did you look at biological activity in Incubation: wet-drying treatments?

5) Page 15, line 13. Can you expand upon the mechanisms suggested by Marcé et al 2019 and relate them to your system?

6) Page 15, line 22 and the rest of this paragraph. I found this paragraph difficult to follow. Does this paragraph only consider what is happening in the top 5 cm of the core? If so, I suggest adding additional columns to Table 1 (this table currently shows that organic carbon is higher in Incubation: wet-drying) to describe the average characteristics of the upper 5 cm. The depth profile (Figure 3) also does not make it clear that wet and wet-drying differ from each other. If the analysis are for the whole core and the table/figure are correct, I do not think the evidence supports your conclusion.

7) Section 4.4 is missing table and figure numbers.

8) Page 17, line 13. In what way is climate not relevant to this study?

---

## Referee Comment (RC3) · Anonymous Referee #3 · 1 Sep 2019

General comments: I think this paper needs serious reframing, as the current narrative about drying is completely unsupported by the data. The title of the paper—that CO2 and CH4 fluxes are decoupled from OC loss—seems appropriate. However, I have serious concerns about the conclusions presented in the abstract and discussion. As I see it, the narrative presented is a) gaseous C fluxes are higher from drying cores; b) OC is lost from drying cores, but at insufficient rates to account for the fluxes, and c) carbonate dissolution may account for the difference. However, only the first point is well-shown and supported. On the second point, the fact that wet-drying and wet incubations saw an identical decrease in sediment OC (as well as the fact that the wet and dry initial cores had identical OC) suggest that drying in and of itself may not actually result in sediment OC loss in situ. The abstract statement, "Organic carbon

content at the end of the incubation was lower in drying cores. . ." is misleading in this sense.

****My most significant concern is related to this issue: if the drying core is missing a C sink (because gaseous C flux < OC loss), then the deficit must be even larger in the wet incubation core, implying there should be even more carbonate dissolution in the wet core. Data for the wet incubation core are selectively not discussed, however. *****

Potential experimental artifacts from the incubation (affecting both wet and wet-drying incubations) might be explored to explain the OC loss. Finally, the mineralogical data, as presented, do not quantitatively show that carbonate dissolution occurred in the wet-drying cores. No mineralogy data or statistics are shown, apart from a PCA in the main manuscript, that suggests (non-quantitatively) carbonate loss. In the supplemental Fig 4, it is not clear that carbonate is actually (statistically) lower in the wet-drying cores. I have not done the math on this, but could you calculate how much C is missing from the gas flux and OC loss balance, and whether that amount could be measured in carbonate loss? Furthermore, if calcite dissolution explained the missing C flux, you might expect to see a relative increase in alkalinity in the wet-drying core compared to the wet core (especially given similar loss in OC), which was not observed. Finally, along the same lines, the discrepancy between the loss of sediment OC and CO2 flux may also be explained to some extent by the decreased alkalinity in both wet treatments (compared to the wet control). I have not done the math on this, but it should be considered.

While the paper's title still seems applicable, the presented narrative about OC loss and carbonate dissolution due to drying is not strongly supported by the data. The same conclusions could be made about the wet core, which overturn the narrative. In my opinion, the abstract and discussion overstep the interpretation of how C processes proceed during in situ drying, and lead to misleading conclusions.

That said, this is an interesting and valuable data set, and is a great example of how

organic and inorganic carbon processes may be important in understanding fluxes. I commend the authors for taking both into account. I think the manuscript may potentially be improved by a) being more upfront about wet and wet-drying OC similarities; b) presenting more mineralogy data (not just the PCA) to show a loss of carbonate (if there is any); (and if not) c) scaling back the conclusions (i.e., by potentially not pointing out carbonate dissolution in the abstract). Again, the discrepancy between fluxes and OC loss is interesting, but–considering the data provided– the rest of the conclusions, especially those about drying specifically, are speculative and should be discussed, but not "concluded."

Specific comments: It needs to be discussed further why the dry core had similar OC to the wet control core, even though the experimental drying process resulted in significant OC loss. Could this be due to potential differences in the sample collection sites (which should be described in more detail)? Perhaps more recent OC accumulation in the dry core site? Just a thought; perhaps there is literature on this. The wet incubations were not stirred (P2, L30) so a significant boundary layer could have built up, preventing diffusion and limiting benthic biogeochemical processes. Can you explain/justify that this was not a significant issue, and/or represents in situ conditions? The presentation of data in the results section is thorough, but could be more streamlined. A lot of info is provided, but it's not always clear what data are important to the argument, so the reader gets lost in the p values. It would be much better to include statistics in a table and present the data in a more clear way. P1, L6: This is not accurate / misleading. OC content was not "lower in the drying core" compared to the wet incubation core. P1, L11: I would say "may be" instead of "was". The data don't show this very conclusively. P3, L2: Can you expand on "periodically" sampled? It looks (from Fig 2) that the wet incubations were only sampled a few times. P3, L9: Can you say where the cores were collected in proximity to each other? I.e., how close? And were they collected randomly? P3, L15: Maybe I missed it, but it seems that the in situ $CO_2$ measurements were never mentioned again / data not shown? P4, L6: Can you say more about the accuracy of the $CO_2$ sensor? Were standards

used? Is the 1% accuracy based on performance in your lab over time? This is very important since your big conclusions are comparing different C measurements taken with different methods. P4, L16: What volume of water was taken for the samples, and how was it removed without introducing an empty headspace? Also, how were the samples stored / what vials / etc? The discussion of headspace equilibration samples is also unclear (and perhaps improved if explained that it's for $CH_4$ in the first sentence). What is the "headspace"? Just ambient air? P5, L30: Do you have a reference for measuring pH and alk on thawed samples? I'm not familiar with this method, so wonder if the freezing/thawing process may affect these. P6, L1: Was mineralogical analysis only done for the incubation treatment samples (and not the controls)? P6, L5: Was aragonite measured? If not, you might want to say why not. While it may not be important in the bedrock, it might form from recent processes? P6, L10: I wonder if frozen/thawed samples give an accurate representation of biological activity? I imagine there is some loss from cell lysis. P6, L11: explain how samples were quantified in 5 mL aliquots. Loosely packed? P6, L12: You might want to explain briefly how the $CO_2$ fluxes were measured. These are not the same cores + chamber lids as for the incubations, so "as described above" isn't accurate. How was the sensor inserted into the airtight seals? How long were the measuremtns taken? P6, L32: Why were data transformed to arcsin/sqrt(x) instead of the straight % abundance? P11: Why not include a table of mineral concentrations, or provide % values for calcite, at the least.

Technical corrections: General observations: *In general, the reference list is a bit short, and I found several of the citations weak. It would be better to have general background info with many references than a few specific examples with single citations (P2 L14-18). A stronger reference list, and more/appropriate in-text citations would be helpful. *References could use some organizing. E.g., Donchyts et al doesn't list the journal name or number (DOI alone is insufficient). Also, in-line citations seem to be out of order in many places. (I assume they should be in alphabetical order). I recommend using a citation management software (e.g., Mendeley) if you don't already. *The writing could be tightened for efficiency and to improve readability. I recommend

flipping through "Writing Science" by Joshua Schimel for guidance on efficient science writing.

Specific observations: Table 1: What is the unit of alkalinity? I don't understand. Figure 2: It's not clear what the different lines (solid, dashed, dotted) correspond to. Figure 3: Are there any points for the "initial" cores? Figure 4: The point sizes in the legend and figure don't match (there are small points on the figure, but not the legend; maybe needs rescaling. . .

*P1, L1: A helpful suggestion: the abstract can be easily tightened for ease of quick reading: As examples: "Meanwhile, reservoirs frequently go dry due to drought and/or water management decisions. Nonetheless, the fate of organic carbon buried in reservoir sediments upon drying is largely unknown." Could be combined into "Reservoirs frequently go dry due to drought and/or water management decisions, and the fate of organic carbon buried upon drying is largely unknown." *P1, L9: You only need to write "mean +/- SE" for the first use. Later, it's implied. *P2, L1: Are you referring to total burial or burial rates? (I assume the latter.) Reservoirs can have similar OC burial rates as coastal wetlands (which are arguably ocean sediments), so it might be good to mention them. Similar comment for the next sentence about eutrophic reservoirs. What is "high" burial? A value might be useful. *P2, L5: Given the ubiquity of reservoirs, "thousands" is insignificant. Consider rephrasing. Also, both the Donchyts and Ragab&Prudhomme references don't seem totally on point; neither reference reservoirs specifically more than a couple of times. They are really about global freshwater trends, which is a different issue. I know there are other references describing annual, if not longer-term drawdowns in reservoirs. *P2, L20: "thought" implies the scientific community thinks this is the case based on evidence. "Commonly assumed" might be a better choice here. *P2, L25: "shifts in the equilibrium" is unneccesarily vague. Why not just say "precipitation or dissolution"? P3, L3: Again, "equilibrium" is a confusing word choice. *P3, L17: Was the 10L water filtered? Clarify. *P3, L23: Missing close parenthesis. *P3, L26: How was the water drained? (Drained through the bottom,

suctioned off top, etc?) *P10, L10: The alkalinity for incubation:wet doesn't seem to be the highest in Table 1. *P15, L3: The entire first paragraph belongs in results, not discussion. *P15, L13: any more references to include here? One reference doesn't seem sufficient to make the conclusion in the next sentence. *P15, L15: This could use more discussion *P15, L22: The wet-drying core appears no different than the wet incubation core in either Table 1 or Fig 3. *P16, L5: You might want to reiterate the "previously discussed" evidence here. It's not clear what you're referring to. *P16, L6: I was waiting for this to be pointed out. :) *P16, L8: But there is not large variability within replicates (i.e., errors are low relative to differences between treatments in Table 1). Thus, this explanation for why the dry control has similarly high OC doesn't make sense. I have to disagree that your data don't actually, "supports the presence of an enhanced mineralization process during drying." *P16, L15: I don't think it's a fair conclusion to say that drying results in an OC loss, because you see the same loss in the wet incubation. *P16, L17: These numbers might be better presented in mmol m-2, so as to match the plots and earlier discussion. It took me some mental gymnastics to realize that you're now referring to the actual amount emitted form the 28 cm2 core surface. *P16, L30, L32: Supplementary figure #s missing. *P17, L1: Did you measure Ca2+? I'm confused. *P18, L8: This could be worded better. Yes, bicarbonate is not a long-term sink (geologically, relative to silicate dissolution). But the bicarbonate wont just "transform to CO2", unless via a) carbonate reprecipitation; or b) entering a low pH environment (wherein it still functions as alkalinity). *P18, L10: The link to forest soils is not clear. We're not talking about carbonate dissolution here, are we?

---

## Author Comment (AC1) · 19 Oct 2019

We sincerely thank this anonymous reviewer for their time and insight. We are confident that their feedback has helped us improve our manuscript during our revisions.

We only studied drying in a single reservoir, so we cannot conclusively determine the effect of reservoir carbonate weathering on global carbon budgets. However, we agree that more context regarding the global carbon budget could help frame our discussion. Thus, we will incorporate the following text into our manuscript.

To be inserted into the last paragraph of Methods- Data Analysis:

"Global scaling of hypothesized inorganic carbon consumption was calculated using satellite-derived estimates of global surface water area loss (Pekel et al. 2016) along

with Global Lakes and Wetlands Database water mass designations (Lehner and Doll 2004) to estimate the global extent of water reservoir drying between 1984 and 2015."

To be inserted into the first paragraph of Discussion- Implications for drying reservoir carbon dynamics:

"However, it is important to note that the likely scale of hypothesized inorganic carbon consumption is small relative to the global freshwater carbon budget. If all drying water reservoirs exhibited the inorganic carbon consumption suggested by "Incubation: Wet-Drying" cores during our 40 day incubation, global inorganic carbon consumption by drying reservoir sediments from 1984 to 2015 would have been 0.311 +/- 0.127 Tg C. This value is 0.2% of estimated inland water carbon burial for just one year (0.15, range 0.06-0.84 Pg year-1) (Mendoca et al. 2017)."

Additional References:

Lehner, B. and P. Döll. "Development and validation of a global database of lakes, reservoirs and wetlands." Journal of Hydrology 296, no. 1-4 (2004): 1-22.

Mendonça, R., R. A. Müller, D. Clow, C. Verpoorter, P. Raymond, L. J. Tranvik, and S. Sobek. "Organic carbon burial in global lakes and reservoirs." Nature communications 8, no. 1 (2017): 1694.

Pekel, J., A. Cottam, N. Gorelick, and A. S. Belward. "High-resolution mapping of global surface water and its long-term changes." Nature 540, no. 7633 (2016): 418.

---

## Author Comment (AC2) · 20 Oct 2019

**Author responses to comments of Anonymous Reviewer #2 regarding Biogeosciences manuscript bg-2019-128 "CO2 and CH4 fluxes are decoupled from organic carbon loss in drying reservoir sediments"**

By Tricia Light, Núria Catalán, Santiago Giralt, and Rafael Marcé

We sincerely thank this anonymous reviewer for their thoughtful and constructive comments. We are confident that their feedback has helped us improve our manuscript during our revisions.

Reviewer comments and our responses are presented below. Reviewer comments are in red, our responses are in black, and direct quotes from our manuscript are in blue. Revisions are italicized.

'CO2 and CH4 fluxes are decoupled from organic carbon loss in drying reservoir sediments" by Light et al describes the contrasting roles of organic carbon metabolism and chemical weathering in reservoir sediments under contrasting hydrological conditions. This manuscript adds to a growing body of work that quantifies the joint effects of organic and inorganic carbon cycles on carbon emissions from freshwater ecosystems. Uniquely, this manuscript describes the contribution of calcium carbonate weathering to carbon emissions in a reservoir experiencing a long-term draw down.

General comments

Greater consistency in naming conventions would improve readability. For example, the names Incubation: drying and Incubation: wet-drying are inconsistently used throughout the paper. Also, using calcium carbonate instead of calcite (if there is a reason to use both terms, an explanation would help!). I appreciate that the authors' address the high variability among replicate cores and that they call for greater spatial sampling across the reservoir. To address the limitation of only sampling within one small region of the reservoir, I would like to see some more information about the location in the reservoir the cores were collected (inflow, transition zone, or outflow). In the Siurana Reservoir is there a transition from more inorganic, watershed derived sediments near the inflow and more organic sediments near the outflow? How does that relate to your findings?

Thank you. We agree that greater consistency would improve readability, and we will revise the manuscript to consistently use treatment labels such as "Incubation: Wet-Drying".

We will also revise our manuscript to clarify our use of both "calcium carbonate" and "calcite". We will modify Page 6 Line 5 to read *"Albite, calcite (a polymorph of calcium carbonate), clinochlorite, dolomite, gypsum, kaolinite, microcline, muscovite, and quartz were identified and quantified."* We will change "calcium carbonate" on Page 16 Line 29 to *"calcite"* because this sentence specifically discusses XRD results, which differentiate between calcite and other calcium carbonate polymorphs. We will add the following text to Discussion Section 4.4 to clarify our later use of calcium carbonate in place of calcite: *"Nearly all calcium carbonate in these sediments is likely to be in the form of calcite, since the Siurana Reservoir's Ca/Mg ratio is too high to promote water column aragonite precipitation. Therefore, we will refer to calcium carbonate for the remainder of this publication."*

Lastly, we will improve our site description by adding the following text to Methods Section 2.1: *"Exposed and submerged sites were as close to one another as possible in the lacustrine zone of the reservoir. The exposed site had sandy sediment and little visible vegetation."* We will discuss the implications of this site selection in the second paragraph of Discussion: Drying sediment carbon loss as follows: "Considering the large variability among replicate cores collected from the same location (which would therefore be more accurately described as pseudo-replicates), greater spatial replication within the reservoir would likely be necessary to resolve differences in sediment carbon content between wet and dry sites. *Wet and dry sites were selected as close to one another as possible to allow for a drought gradient while minimizing differences in organic matter composition, but lacustrine environments can display significant spatial heterogeneity in organic matter even over short distances (Cardoso-Silva et al. 2018; Downing and Rathe 1988; Mackay et al. 2012; Pittman et al. 2013).*"

Specific comments
1) Page 2, line 24. Can you expand on the environmental conditions in which the equilibrium reaction of calcium is important and how those conditions relate to the environmental conditions in reservoirs?
Thank you. We will add the following on Page 2 Line 30: *"Additionally, shifts in the calcium carbonate equilibrium can be induced by changes in sediment moisture content, temperature, and oxygen availability, all of which are expected to occur during water reservoir drying."*

2) Page 3, line 10. How certain are you that the wet sites have been consistently wet for the last 2 years? How would intermittent drying affect your results?
The reservoir water level at the time of sampling was the lowest it had been in more than three years, according to data from the Spanish Ministry of Agriculture, Food, and Environment (https://sig.mapama.gob.es/93/ClienteWS/redes-seguimiento/default.aspx?origen=1008&nombre=ROAN_ESTACION_AFORO_EMBALSES&claves=COD_HIDRO%7CCOD_SITUACION_ESTACION&valores=9868%7C2, accessed 1-10-19). Thus, while intermittent drying might have affected our results, we can be very certain that our submerged site has been consistently wet for at least two years and no intermittent drying has occurred.

3) Figure 1. Are the DIC-method and flux methods generally consistent for the Incubation: Wet treatments?
DIC measurements were conducted in solution while flux measurements were conducted in gas phase, so we believe grouping these two distinct analyses as one in this figure might be misleading.

4) Page 6, line 10. What are your findings? Did you look at biological activity in Incubation: wet-drying treatments?
Thank you. We will revise our results on Page 7 Line 12 to clarify that carbon dioxide uptake increased after sterilization. We will also note that the following sterilization experiment data is in Table S7 of our supplemental:

|  | Pre-Sterilization $CO_2$ Flux (ppm $CO_2$ s$^{-1}$) | Post-Sterilization $CO_2$ Flux (ppm $CO_2$ s$^{-1}$) |
| --- | --- | --- |
| Replicate 1 | -0.00769 | -0.03729 |
| Replicate 2 | -0.00308 | -0.03750 |
| Replicate 3 | -0.00500 | -0.03974 |

We did not look at biological activity in the Incubation: Wet-Drying treatment. We certainly should make this more explicit, so we will modify Page 7 Line 11 as follows: "Post-incubation analysis of sediment *from a randomly select "Incubation: Dry" core* showed that $CO_2$ influx to the sediment increased after sediment sterilization (Table S7)."

5) Page 15, line 13. Can you expand upon the mechanisms suggested by Marcé et al 2019 and relate them to your system?
Yes, we will add the following to this section: *"Marcé et al. 2019 suggested that variability is driven by variations in moisture, temperature, distance from water-line, soil type, topography and organic carbon as well as their interactions. Thus, fluxes measured in different reservoirs could strongly vary as a function of different catchment geology and productivity."*

6) Page 15, line 22 and the rest of this paragraph. I found this paragraph difficult to follow. Does this paragraph only consider what is happening in the top 5 cm of the core? If so, I suggest adding additional columns to Table 1 (this table currently shows that organic carbon is higher in Incubation: wet-drying) to describe the average characteristics of the upper 5 cm. The depth profile (Figure 3) also does not make it clear that wet and wet-drying differ from each other. If the analysis are for the whole core and the table/figure are correct, I do not think the evidence supports your conclusion.
Thank you for this feedback. Yes, this paragraph is only referring to what happened in the top 5 cm of the core. We will clarify this by changing the second sentence to read "*Both "Incubation: Wet" and "Incubation: Wet-Drying" cores displayed lower organic carbon content in the upper 5 cm than "Initial: Wet" cores (Table 1, Fig. 3), which is consistent with organic carbon decomposition during the incubation.*" We will also add the following row to Table 1 displaying organic carbon content for the top 5 cm of all cores:

|  | Initial: Control Wet | Initial: Control Dry | Incubation: Wet | Incubation: Dry | Incubation: Wet-Drying |
| --- | --- | --- | --- | --- | --- |
| *Top 5 cm Organic Carbon Content (g/gdw %)* | *4.28 ± 0.26* | *3.41 ± 0.16* | *2.81 ± 0.21* | *3.72 ± 0.07* | *2.86 ± 0.33* |

7) Section 4.4 is missing table and figure numbers.
We apologize for this oversight. We will correct the missing references to reflect that mineralogy data is in the Supplement Table S10.

8) Page 17, line 13. In what way is climate not relevant to this study?
Thank you, we agree that this sentence is poorly phrased. We intended to refer to the increase in precipitation observed in that study, so we will change "climate" to "*increased precipitation*".

---

## Author Comment (AC3) · 20 Oct 2019

**Author responses to comments of Anonymous Reviewer #3 regarding Biogeosciences manuscript bg-2019-128 "CO2 and CH4 fluxes are decoupled from organic carbon loss in drying reservoir sediments"**

By Tricia Light, Núria Catalán, Santiago Giralt, and Rafael Marcé

We sincerely thank this anonymous reviewer for their extremely thoughtful and detailed consideration of our manuscript. Their insights were extremely valuable, and we are confident that their feedback has helped us greatly improve our manuscript during our revisions.

Reviewer comments and our responses are presented below. Reviewer comments are in red, our responses are in black, and direct quotes from our manuscript are in blue. Revisions are italicized.

General comments:
I think this paper needs serious reframing, as the current narrative about drying is completely unsupported by the data. The title of the paper that CO2 and CH4 fluxes are decoupled from OC loss seems appropriate. However, I have serious concerns about the conclusions presented in the abstract and discussion. As I see it, the narrative presented is a) gaseous C fluxes are higher from drying cores; b) OC is lost from drying cores, but at insufficient rates to account for the fluxes, and c) carbonate dissolution may account for the difference. However, only the first point is well-shown and supported. On the second point, the fact that wet-drying and wet incubations saw an identical decrease in sediment OC (as well as the fact that the wet and dry initial cores had identical OC) suggest that drying in and of itself may not actually result in sediment OC loss in situ. The abstract statement, "Organic carbon content at the end of the incubation was lower in drying cores. . ." is misleading in this sense.
****My most significant concern is related to this issue: if the drying core is missing a C sink (because gaseous C flux < OC loss), then the deficit must be even larger in the wet incubation core, implying there should be even more carbonate dissolution in the wet core. Data for the wet incubation core are selectively not discussed, however. *****
Potential experimental artifacts from the incubation (affecting both wet and wet-drying incubations) might be explored to explain the OC loss. Finally, the mineralogical data, as presented, do not quantitatively show that carbonate dissolution occurred in the wetdrying cores. No mineralogy data or statistics are shown, apart from a PCA in the main manuscript, that suggests (non-quantitatively) carbonate loss. In the supplemental Fig 4, it is not clear that carbonate is actually (statistically) lower in the wet-drying cores. I have not done the math on this, but could you calculate how much C is missing from the gas flux and OC loss balance, and whether that amount could be measured in carbonate loss?
Furthermore, if calcite dissolution explained the missing C flux, you might expect to see a relative increase in alkalinity in the wet-drying core compared to the wet core (especially given similar loss in OC), which was not observed. Finally, along the same lines, the discrepancy between the loss of sediment OC and CO2 flux may also be explained to some extent by the decreased alkalinity in both wet treatments (compared to the wet control). I have not done the math on this, but it should be considered.
While the paper's title still seems applicable, the presented narrative about OC loss and carbonate dissolution due to drying is not strongly supported by the data. The same conclusions could be made about the wet core, which overturn the narrative. In my opinion, the abstract and discussion overstep the interpretation of how C processes proceed during in situ drying, and lead

to misleading conclusions. That said, this is an interesting and valuable data set, and is a great example of how organic and inorganic carbon processes may be important in understanding fluxes. I commend the authors for taking both into account. I think the manuscript may potentially be improved by a) being more upfront about wet and wet-drying OC similarities; b) presenting more mineralogy data (not just the PCA) to show a loss of carbonate (if there is any); (and if not) c) scaling back the conclusions (i.e., by potentially not pointing out carbonate dissolution in the abstract). Again, the discrepancy between fluxes and OC loss is interesting, but–considering the data provided– the rest of the conclusions, especially those about drying specifically, are speculative and should be discussed, but not "concluded."

We are extremely grateful for this reviewer's many insights. We appreciate their belief that our dataset is valuable and interesting, and we completely agree that our manuscript should be reframed in order to clarify that our conclusions are well-supported by our data. We will discuss each conceptual concern the reviewer raises individually, and then discuss the implications for our overall narrative.

First, we agree that organic carbon loss in the "Incubation: Wet" treatment warrants more discussion in our manuscript. We certainly should clarify that organic carbon loss relative to "Initial: Wet" cores was observed for both "Incubation: Wet" and "Incubation: Wet-drying" treatments, so this discrepancy cannot be attributed to drying. However, we believe that methodological limitations of the "Incubation: Wet" treatment make it an imperfect control by which to measure organic carbon loss during drying. "Incubation: Wet" cores likely represented a greater departure from in situ conditions than "Incubation: Dry" and "Incubation: Drying" cores, as we will discuss in Discussion Section 4.2 as follows:

*"Sediment organic carbon data suggests that the observed $CO_2$ efflux was caused by organic matter decomposition in both "Incubation: Wet" and "Incubation: Wet-Drying" treatments, but "Incubation: Wet" may be affected by a greater discrepancy from in-situ conditions than "Incubation-Drying". Both "Incubation: Wet" and "Incubation: Wet-Drying" cores displayed lower organic carbon content than "Initial: Wet" cores (Table 1, Fig. 3), which is consistent with organic carbon decomposition during the incubation. Organic carbon loss in both treatments was similar in magnitude, but two factors suggest that the in-situ significance of these losses differs with treatment. First, "Incubation: Wet" cores were submerged under only 10 cm of filtered reservoir water and deprived of in situ organic matter deposition from the water column. Thus, the observed organic carbon loss may represent the continuation of organic carbon respiration similar to that which would have occurred in situ without the accompanying in situ organic carbon replacement. In contrast, the relatively static system of naturally drying sediment is likely better represented by our experimental set-up. Second, the incubation of all sediment cores at a constant 25 C likely resulted in a greater discrepancy from in situ conditions for "Incubation: Wet" cores than for "Incubation: Wet-Drying". In light of these factors, the "Incubation: Wet" treatment is an imperfect control by which to quantify organic carbon decomposition specifically caused by reservoir drying."*

With these complications in mind, we believe that the difference in organic carbon content between "Initial: Wet" and "Incubation: Wet-Drying" cores is a more reasonable estimate for in situ organic carbon loss during drying than comparison of "Incubation: Wet" and "Incubation: Wet-Drying" cores. We will modify our discussion of organic carbon loss in Discussion Section 4.2 to clarify this comparison:

*"The difference in organic carbon content between "Initial: Wet" and "Incubation: Wet-Drying" is a useful estimate for organic carbon loss during drying. If the difference in organic carbon*

*observed between these two treatments is representative of in situ organic carbon loss during drying, this discrepancy* has significant implications for the reservoir's carbon budget."

Second, we completely agree that we should discuss that organic carbon loss was decoupled from carbon gas fluxes in both "Incubation: Wet" and "Incubation: Wet-Drying" cores. Again, we believe that methodological constraints in the "Incubation: Wet" treatment may make it an imperfect comparison for the "Incubation: Wet-Drying" treatment. "Incubation: Wet" $CO_2$ fluxes were measured with much lower temporal resolution than "Incubation: Wet-Drying fluxes, so our mass balance approach is much more reliable for the "Incubation: Wet-Drying" treatment. Thus, the significance of the decoupling between carbon gas fluxes and organic carbon loss may differ by treatment. This is an important caveat to our discussion of inorganic carbon consumption in "Incubation: Wet" and "Incubation: Wet-Drying" treatments and certainly warrants discussion. We will therefore replace the entirety of Discussion Section 4.3 with the following:

*"The observed organic carbon loss from "Incubation: Wet-Drying" and "Incubation: Wet" cores was consistent with the observed carbon gas fluxes in direction, but in both cases organic carbon loss exceeded the amount of carbon gas emitted. The differences in sediment organic carbon content in the surface layer (0-5 cm) of "Incubation: Wet-Drying" and "Incubation: Wet" cores relative to "Initial: Wet" cores correspond to net effluxes of 568 +/- 247.2 and 588.4 +/- 157.3 mmol C m$^{-2}$ day$^{-1}$ respectively, but observed net efflux was only 206.7 +/- 47.9 and 69.2 +/- 18.1 mmol C m$^{-2}$ day$^{-1}$. Thus, even considering the aforementioned variability in sediment organic carbon data, it appears that a significant portion of the organic carbon consumed via decomposition was not emitted as $CO_2$ but rather consumed by one or more other processes.* This is also supported by the observed influx of $CO_2$ from the atmosphere into the sediment for "Incubation: Dry" treatment cores. Consistent $CO_2$ influxes in "Incubation: Dry" cores similar in magnitude to the $CO_2$ effluxes in "Incubation: Wet-Drying" cores were observed after one week of incubation across all replicates (Table 1, Fig. 2). Furthermore, by the end of the incubation two out of three replicate "Incubation: Wet-Drying" cores also displayed $CO_2$ influxes. These findings show the relevance of the $CO_2$ consumption pathway(s) active in these sediments. They also imply that observed $CO_2$ effluxes must be considered the net result of $CO_2$ production and consumption processes.

*Multiple factors might contribute to the decoupling between organic carbon loss and carbon gas emissions. It may be the result of methodological constraints, particularly in the "Incubation-Wet" treatment. The overlaying water in the "Incubation: Wet" treatment prevented the reliable direct measurement of carbon gas fluxes from the sediment. Instead, carbon gas flux was determined by measuring change in water column DIC concentrations over a 10 minute window four times throughout the 45 day incubation period. This poor temporal resolution introduces the possibility of sporadic carbon emissions from the sediment that avoided observation. Moreover, while dissolved organic carbon concentration change in the water column during these 10 minute intervals was minimal, all water samples were filtered prior to analysis and particulate organic carbon flux was not analyzed. These factors make it difficult to construct a mass balance for "Incubation: Wet" cores with confidence, and there are no obvious non-methodological explanations for the decoupling of organic carbon loss and carbon gas fluxes for that treatment. In contrast, the monitoring of carbon gas fluxes for "Incubation: Dry" and "Incubation: Wet-Drying" cores was comparatively straight-forward. Along with the $CO_2$ influxes observed in "Incubation: Dry" cores, this suggests that $CO_2$ consumption was occurring in the exposed incubation sediments.*

We will also replace the first sentence of Discussion Section 4.4 as follows:

*"Sediment mineralogy results suggest that the observed sediment carbon consumption in "Incubation: Wet-Drying" and "Incubation: Dry" cores may have been caused by an increase in calcium carbonate chemical weathering relative to precipitation (Reaction R1)."*

Third, we agree that only presenting mineralogy data as a PCA was a significant omission on our part. We apologize for this oversight, and we understand how the lack of clearly presented statistical data caused the reviewer to perceive a lack of support for our findings. A one-way ANOVA showed that calcite abundance is statistically different between treatments (p<0.0001), with the "Incubation: Wet" treatment differing from both "Incubation: Dry" (p<0.0001) and "Incubation: Wet-drying" treatments (p = 0.019) in post hoc Tukey's tests. This evidence is essential to supporting our conclusion regarding calcium carbonate chemical weathering. Thus, we will reference it in Results Section 3.3 and Discussion Section 4.4 and present it along with our other statistics in Table 2 (see the newly created Table 2 in the Specific Comments section below). We will also add calcite abundance data to Table 1, as follows:

| | *Incubation: Wet* | *Incubation: Dry* | *Incubation: Wet-Drying* |
|---|---|---|---|
| *Calcite Content (g/gdw %)* | *29.57 $\pm$ 0.65* | *24.40 $\pm$ 0.85* | *25.88 $\pm$ 1.05* |

Fourth, we have performed the carbonate loss calculation this reviewer suggested, and the average percent calcium carbonate difference between "Incubation: Wet" and "Incubation: Wet-Drying" cores corresponded to a $CO_2$ sink of 259.9 $\pm$ 283.8 mmol (mean $\pm$ SE). This range does include the discrepancy between OC carbon loss and carbon gas flux for "Incubation: Wet: Drying" cores (46.0 $\pm$ 24.0 mmol), but given the large uncertainty in this estimate we believe that this calculation has little meaning. We will modify Page 16 Line 30 as follows to reflect that higher experimental replication is likely necessary to close this mass balance due to high compositional variability at the study site:
*"This suggests a relative increase in calcium carbonate chemical weathering in dry conditions, but large variability in sediment composition at this study site likely requires higher experimental replication in order to close the mass balance for the drying sediment core system."*

Fifth, we agree that sediment alkalinity warrants more discussion in our manuscript. While the lack of an increase in alkalinity in our "Incubation: Wet-Drying" treatments is interesting, given the logistical difficulty of analyzing alkalinity of dry sediment (i.e. Mashhady and Rowell 1978; Moore 2001; Suarez and Rhoades 1982; Sumner 1994) we believe that our alkalinity data is inconclusive. As we discuss in our Methods, we followed the standard procedure of suspending sediment in deionized water, stirring, filtering the sediment-water solution, and analyzing the alkalinity of the supernatant. However, since the "Incubation: Wet" sediment was already saturated but "incubation: Dry" and "Incubation: Wet-Drying" sediment was not, we believe that solubility effects may impede the direct comparison of alkalinity between treatments. We will discuss this in our manuscript by adding the following to Discussion Section 4.4:
*"Regardless, it is important to note that mineralogy data is inconclusive. Calcium carbonate chemical weathering is also expected to raise sediment alkalinity, but alkalinity was higher in "Initial: Control Wet" and "Incubation: Wet" treatments than "Incubation: Dry" and "Incubation: Wet-Drying" treatments. However, it is possible that the slurry method used to*

*analyze alkalinity in this investigation impedes the reliable direct comparison between wet and dry sediments due to solubility effects."*

Together, we believe that these revisions address the reviewer's concerns and help clarify the limitations of this study. In light of the additional data we present here, we believe that our manuscript's main narratives are well-founded. However, we completely agree that we should reframe our abstract and conclusions to be more upfront regarding 1) the similarity between "Incubation: Wet" and "Incubation: Wet-Drying" treatments and 2) the preliminary nature of our findings of calcium carbonate chemical weathering in exposed sediments. We have revised our Abstract and Conclusions accordingly:

"Meanwhile, reservoirs frequently go dry due to drought and/or water management decisions. Nonetheless, the fate of organic carbon buried in reservoir sediments upon drying is largely unknown." Could be combined into "Reservoirs frequently go dry due to drought and/or water management decisions, and the fate of organic carbon buried upon drying is largely unknown."

Abstract:
"Reservoirs are a prominent feature of the current global hydrological landscape, and their sediments are the site of extensive organic carbon burial. *Reservoirs frequently go dry due to drought and/or water management decisions, and the fate of buried organic carbon upon drying is largely unknown.* Here, we conducted a 45-day-long laboratory incubation of sediment cores collected from a western Mediterranean reservoir to investigate carbon dynamics in drying sediment. *Sediment incubated with overlaying water ("Incubation: Wet" cores) and without overlaying water ("Incubation: Wet-drying" cores) both emitted $CO_2$ during the incubation, but emissions were greater in "Incubation: Wet-drying" cores (206.7 +/- 47.9 vs. 69.2 +/- 18.1 mmol C $m^{-2}$ $day^{-1}$ , mean +/- SE). Both "Incubation: Wet" and "Incubation: Wet-drying" cores exhibited lower organic carbon content than "Initial: Wet" reservoir cores analyzed immediately following core collection, which suggests that this $CO_2$ efflux was due to organic carbon mineralization. However, the apparent rate of organic C reduction in incubation sediments was higher than C emission for both treatments (588.4 +/- 157.3 and 568.6 +/- 247.2 mmol C $m^{-2}$ $day^{-1}$ for "Incubation: Wet" and "Incubation: Wet-drying" cores respectively). Meanwhile, sediment cores collected from a reservoir area that had already been exposed for 2+ years ("Incubation: Dry") displayed net $CO_2$ influx from the atmosphere to the sediment (-136.0 +/- 27.5 mmol C $m^{-2}$ $day^{-1}$) during the incubation period, and two out of three "Incubation: Wet-drying" cores also displayed $CO_2$ influx by the end of the incubation. Quantitative and qualitative mineralogical analyses suggest that a relative increase in calcium carbonate chemical weathering may have caused both the decoupling of organic carbon loss from carbon gas fluxes in "Incubation: Wet-drying" sediments as well as the observed $CO_2$ influx to "Incubation: Dry" sediments. Thus, while organic carbon decomposition in newly dry reservoir sediment may cause measurable organic carbon loss and carbon gas emissions to the atmosphere, other processes may offset these emissions on short time frames and compromise the use of carbon emissions as a proxy for organic carbon mineralization in drying sediments.*"

Conclusions:
"This investigation used a laboratory sediment core incubation to explore the effects of reservoir drying on sediment carbon dynamics. We directly linked organic carbon loss to carbon dioxide emissions in drying reservoir sediment for the first time, undermining the idea that organic carbon burial in active reservoir sediments represents a long-term carbon sink. *In addition, we*

*found a decoupling between carbon loss and carbon gas fluxes. Mineralogical sediment composition suggests that the discrepancy in "Incubation: Wet-drying" sediment cores was due to an increase in calcium carbonate chemical weathering. Calcium carbonate chemical weathering likely also drove carbon dioxide influxes to "Incubation: Drying" and "Incubation: Wet-Drying" cores. Together, these findings show that while reservoir sediment drying can cause organic carbon decomposition and thus carbon gas efflux to the atmosphere, other sediment processes can potentially offset or even reverse these fluxes."*

Specific comments:
It needs to be discussed further why the dry core had similar OC to the wet control core, even though the experimental drying process resulted in significant OC loss. Could this be due to potential differences in the sample collection sites (which should be described in more detail)? Perhaps more recent OC accumulation in the dry core site? Just a thought; perhaps there is literature on this.

Thank you. We agree that we should discuss both our sample collection sites and the relatively high OC of the dry cores in more detail. We selected sample collection collection sites in order to minimize differences in organic matter composition independent of drying, but the relatively high OC content of the dry cores likely shows that this effort was unsuccessful. Sediment at the dry site likely contained significantly more OC when it was still submerged than the wet site contained at the time of sampling. This possibility is supported by the literature on organic matter spatial heterogeneity in lake and reservoir sediments (Cardoso-Silva et al. 2018; Downing and Rathe 1988; Mackay et al. 2012; Pittman et al. 2013). We believe that recent (i.e. post-drying) OC accumulation at the dry core site is very unlikely given that the area displayed little vegetation. We will make the following revisions to the manuscript to clarify this narrative and incorporate more support from the literature.

Add to Methods Section 2.1: *"Exposed and submerged sites were as close to one another as possible in the lacustrine zone of the reservoir. The exposed site had sandy sediment and little visible vegetation."*

Add to Discussion Section 4.2: *"Considering the large variability among replicate cores collected from the same location (which would therefore be more accurately described as pseudo-replicates), greater spatial replication within the reservoir would likely be necessary to resolve differences in sediment carbon content between wet and dry sites. Wet and dry sites were selected as close to one another as possible to allow for a drought gradient while minimizing differences in organic matter composition, but lacustrine environments can display significant spatial heterogeneity in organic matter even over short distances (Cardoso-Silva et al. 2018; Downing and Rathe 1988; Mackay et al. 2012; Pittman et al. 2013)."*

The wet incubations were not stirred (P2, L30) so a significant boundary layer could have built up, preventing diffusion and limiting benthic biogeochemical processes. Can you explain/justify that this was not a significant issue, and/or represents in situ conditions?

We will revise Methods to reflect that the Incubation: Wet cores were stirred approximately every other day. We are not sure what the reviewer is referring to in P2, L30, since this line does not relate to stirring and nowhere in the manuscript do we state that the wet incubations were not stirred.

The presentation of data in the results section is thorough, but could be more streamlined. A lot of info is provided, but it's not always clear what data are important to the argument, so the reader gets lost in the p values. It would be much better to include statistics in a table and present the data in a more clear way.

We appreciate the feedback. We will add the following statistics table and streamline our Results section, referring to this table rather than including test values in the main text of our manuscript.

Table 2. Results of statistical analyses comparing "Initial: Wet", "initial: Dry", "Incubation: Wet", "Incubation: Dry", and "Incubation: Wet-Drying" treatments. Statistically significant differences ($p<0.05$) are bolded.

| | Pearson Correlation Tests | | | | | |
|---|---|---|---|---|---|---|
| | $CO_2$ flux | | Mineralogy PCA axis #1 | | Mineralogy PCA axis #2 | |
| | p-value | $r^2$ | p-value | $r^2$ | p-value | $r^2$ |
| Change in core mass | **0.0040** | **0.27** | 0.48 | 0.074 | 0.64 | 0.032 |
| OC content | 0.25 | 0.18 | **0.0011** | **0.14** | **0.0015** | **0.13** |
| Water content | NA | | **<0.0001** | **0.29** | 0.92 | 0.00016 |
| $CO_2$ flux | NA | | 0.16 | 0.26 | 0.55 | 0.054 |
| | **ANOVAs** | | | | | |
| Comparison | | | | | | |
| OC by treatment | 8.52 | **p <0.0001** | | | | |
| OC by depth layer | 5.77 | **0.004** | | | | |
| OC by treatment (0-5 cm) | 6.32 | **0.002** | | | | |
| OC by treatment (5-9 cm) | 2.32 | 0.09 | | | | |
| OC by treatment (9-15 cm) | 6.89 | **0.0004** | | | | |
| Alkalinity by treatment | 24.96 | **p <0.0001** | | | | |
| Calcite abundance by treatment | 12.7 | **p <0.0001** | | | | |

| | Post-hoc Tukey tests | | | |
|---|---|---|---|---|
| OC content by treatment (0-5 cm) | | | | |
| | Initial: Dry | Incubation: Wet | Incubation: Dry | Incubation: Wet-Drying |
| Initial: Wet | p = 0.056 | **p = 0.0017** | p = 0.68 | **p <0.0001** |
| Initial: Dry | | p = 1.00 | p = 1.00 | p = 0.70 |
| Incubation: Wet | | | p = 0.44 | p = 1.00 |
| Incubation: Dry | | | | p = 0.071 |
| OC by treatment (9-15 cm) | | | | |
| | Initial: Dry | Incubation: Wet | Incubation: Dry | Incubation: Wet-Drying |
| Initial: Wet | **p = 0.0019** | p = 1.00 | p = 1.00 | p = 1.00 |
| Initial: Dry | | **p <0.0001** | p = 0.68 | **p = 0.0010** |
| Incubation: Wet | | | p = 0.13 | p = 1.00 |
| Incubation: Dry | | | | p = 0.88 |
| Alkalinity by treatment | | | | |
| | Initial: Dry | Incubation: Wet | Incubation: Dry | Incubation: Wet-Drying |
| Initial: Wet | **p = 0.0002** | **p = 0.0050** | **p = 0.0001** | **p <0.0001** |
| Initial: Dry | | p = 0.17 | p = 0.73 | p = 0.91 |
| Incubation: Wet | | | p = 0.25 | **p = 0.047** |
| Incubation: Dry | | | | p = 0.99 |
| Calcite abundance by treatment | | | | |
| | Incubation: Dry | Incubation: Wet-Drying | | |
| Incubation: Wet | **p <0.0001** | **p = 0.019** | | |
| Incubation: Dry | | p = 0.068 | | |

P1, L6: This is not accurate / misleading. OC content was not "lower in the drying core" compared to the wet incubation core.

Thank you. We will change this to "*Both "Incubation: Wet" and "Incubation: Wet-drying" cores exhibited lower organic carbon content than "Initial: Wet" reservoir cores analyzed immediately following core collection*" to clarify our intended comparison.

We also note that there was a mistake in our initial manuscript on Page 10 Line 5. This sentence should read: "In the top layer (0-5 cm), "Initial: Wet" cores displayed higher organic carbon content than "Incubation: Wet" and "Incubation: Wet-Drying" cores (Table 2)."

P1, L11: I would say "may be" instead of "was". The data don't show this very conclusively.

Thank you, we will make this change.

P3, L2: Can you expand on "periodically" sampled? It looks (from Fig 2) that the wet incubations were only sampled a few times.

We will add that carbon gas fluxes from "'Incubation: Wet" cores were measured 4 times during the 45 day incubation.

P3, L9: Can you say where the cores were collected in proximity to each other? I.e., how close? And were they collected randomly?

We will add that cores were randomly collected within 10 meters of each other within each location.

P3, L15: Maybe I missed it, but it seems that the in situ CO2 measurements were never mentioned again / data not shown?

We reference that this data is in the publication's supplemental in Table S1 (data also provided below). We agree that we should also mention it in our text, so we will add to Results Section 3.1: *"In situ fluxes from the dry location at the time of sampling were highly variable but comparable in magnitude to those from "Incubation: Dry" cores at the beginning of the incubation (Table S1)."*

Table S1: In situ $CO_2$ fluxes from dry and wet study sites at the time of sediment core collection.

| Dry Site Emission (mmol $CO_2$ m$^{-2}$ d$^{-1}$) | |
|---|---|
| E1 | 396.96 |
| E2 | 73.52 |
| E3 | 38.16 |
| E4 | 55.35 |
| E5 | 36.27 |
| Wet Site Emission (mmol $CO_2$ m$^{-2}$ d$^{-1}$) | |
| S1 | 353.03 |
| S2 | 125.21 |
| S3 | 83.38 |
| S4 | 13.58 |
| S5 | 12.18 |

P4, L6: Can you say more about the accuracy of the CO2 sensor? Were standards used? Is the 1% accuracy based on performance in your lab over time? This is very important since your big conclusions are comparing different C measurements taken with different methods.
According to its user manual, the EGM-4 Environmental Gas Monitor used exhibits accuracy < 1 1% over the calibrated $CO_2$ range. This accuracy is maintained through frequent autozeroing cycles using drierite as a $CO_2$ trap. We have checked this accuracy several times during recents years using standard $CO_2$ gas samples and by intercalibrating different EGM-4 Environmental Gas Monitors with atmospheric air. Accuracy during these checks was always within the 1% accuracy suggested by the manual.

P4, L16: What volume of water was taken for the samples, and how was it removed without introducing an empty headspace? Also, how were the samples stored / what vials / etc?
We will add to Methods Section 2.3: *"25 ml of water was removed for water analysis, and this water was replaced with filtered reservoir water to maintain a constant water level. Water was transferred to 25 ml acid washed PPC bottles with no headspace, refrigerated, and analyzed within 24 hours of sampling."*

The discussion of headspace equilibration samples is also unclear (and perhaps improved if explained that it's for CH4 in the first sentence). What is the "headspace"? Just ambient air? Thank you. We will state that it was for CH4 in the first sentence and clarify that the headspace was ambient air enclosed by the core cap.

P5, L30: Do you have a reference for measuring pH and alk on thawed samples? I'm not familiar with this method, so wonder if the freezing/thawing process may affect these.

According to The Handbook of Techniques for Aquatic Sediments Sampling (Mudroch and MacKnight 1994), freezing of wet sediments is unlikely to affect these analyses. However, as we discussed above, our alkalinity measurements are likely inconclusive due to the difficulty of directly comparing wet and dry sediments (i.e. Mashhady and Rowell 1978; Moore 2001; Suarez and Rhoades 1982; Sumner 1994). We followed the standard procedure of suspending sediment in deionized water, stirring, filtering the sediment-water solution, and analyzing the alkalinity of the supernatant but since the "Incubation: Wet" sediment was already saturated, we believe that solubility effects may impede the direct comparison of alkalinity between treatments.

P6, L1: Was mineralogical analysis only done for the incubation treatment samples (and not the controls)?

Yes, we will clarify that mineralogical analysis was performed in all three incubation cores but not on the "Initial: Control Wet" and "Initial: Control Dry" cores.

P6, L5: Was aragonite measured? If not, you might want to say why not. While it may not be important in the bedrock, it might form from recent processes?

Thank you. Aragonite was not measured because the Siurana Reservoir's Ca/Mg ratio is too high to promote water column aragonite precipitation. We will clarify this by adding to Discussion Section 4.4: *"Nearly all calcium carbonate in these sediments is likely to be in the form of calcite, since the Siurana Reservoir's Ca/Mg ratio is too high to promote water column aragonite precipitation."*

P6, L10: I wonder if frozen/thawed samples give an accurate representation of biological activity? I imagine there is some loss from cell lysis.

We agree that freezing sediment may decrease biological activity. Since our intention in defrosting and sterilizing this sediment was to rule out biological activity as the cause for $CO_2$ influx to sediments, we do not believe this presents any issue (i.e. if anything, cell lysis would have been expected to increase respiration). Even defrosted sediment displayed $CO_2$ uptake.

P6, L11: explain how samples were quantified in 5 mL aliquots. Loosely packed?

Thank you. We will add that sediment was loosely packed.

P6, L12: You might want to explain briefly how the CO2 fluxes were measured. These are not the same cores + chamber lids as for the incubations, so "as described above" isn't accurate. How was the sensor inserted into the airtight seals? How long were the measurements taken?

Thank you. We will elaborate on this section as follows: *"This sediment sample was split into 3 replicate sections of 5 ml each, each of which was placed in a separate 100 ml glass beaker and covered with one of the custom made caps used to cover the incubation cores during gas flux analysis. The caps were taped in place to form an air-tight seal and create a closed chamber with a volume of 100 cm³. These chambers were connected to an environmental gas monitor and $CO_2$ concentrations within the chambers were recorded as described above (measurements every 4.8 seconds, with measurements lasting 300 seconds or until a 10 μatm change in $CO_2$ was recorded, whichever came first)."*

P6, L32: Why were data transformed to arcsin/sqrt(x) instead of the straight % abundance?

This is a standard transformation for statistical analysis of abundance data, since the number of zeros in straight % abundance data makes data non-normal (Crawley 1988).

P11: Why not include a table of mineral concentrations, or provide % values for calcite, at the least.
Thank you for the suggestion. We will add a row with the following average calcite abundances to Table 1. We believe that a full table with all mineral percentages would be of little value to readers, so we will keep this information in Supplemental Table S10.

| | Incubation: Wet | Incubation: Dry | Incubation: Wet-Drying |
|---|---|---|---|
| Calcite Content (g/gdw %) | $29.57 \pm 0.65$ | $24.40 \pm 0.85$ | $25.88 \pm 1.05$ |

Technical corrections: General observations:
*In general, the reference list is a bit short, and I found several of the citations weak. It would be better to have general background info with many references than a few specific examples with single citations (P2 L14-18). A stronger reference list, and more/appropriate in-text citations would be helpful.
We will follow this advice and improve our reference list and citations whenever possible, starting with the additional references cited in this response. However, research on carbon gas emissions from drying reservoir sediment is scarce, so at times we must offer specific examples rather than general background information. We will modify P2 L14-18 as follows to clarify this:
"*Literature on carbon gas fluxes from drying water reservoir sediments is sparse, but recent investigations have suggested that drying water reservoir sediments are associated with significant carbon emissions.* An investigation of reservoir sediments during an extreme drought event in South Korea documented pulses of carbon dioxide emissions upon drying large enough to counteract years of preceding carbon burial (Jin et al., 2016). Furthermore, an incubation experiment that dried sediment cores collected from a Brazilian reservoir showed substantial $CO_2$ and $CH_4$ emissions during drying and a significant effect of recurrent atmospheric exposure on sediment carbon fluxes (Kosten et al., 2018)."

*References could use some organizing. E.g., Donchyts et al doesn't list the journal name or number (DOI alone is insufficient). Also, in-line citations seem to be out of order in many places. (I assume they should be in alphabetical order). I recommend using a citation management software (e.g., Mendeley) if you don't already.
Thank you, we will fix these errors.

*The writing could be tightened for efficiency and to improve readability. I recommend flipping through "Writing Science" by Joshua Schimel for guidance on efficient science writing.
Thank you, we will revise for readability.

Specific observations:
Table 1: What is the unit of alkalinity? I don't understand.
$\mu$M $HCO_3^-$. Thank you, we will fix the typo.

Figure 2: It's not clear what the different lines (solid, dashed, dotted) correspond to.

Thank you, we will add a legend to clarify that the different lines correspond to different replicate cores.

Figure 3: Are there any points for the "initial" cores?
Yes. We will revise the figure to more clearly distinguish between Initial and Incubation cores.

Figure 4: The point sizes in the legend and figure don't match (there are small points on the figure, but not the legend; maybe needs rescaling. . .
Thank you, we will fix the rescaling issue.

*P1, L1: A helpful suggestion: the abstract can be easily tightened for ease of quick reading: As examples: "Meanwhile, reservoirs frequently go dry due to drought and/or water management decisions. Nonetheless, the fate of organic carbon buried in reservoir sediments upon drying is largely unknown." Could be combined into "Reservoirs frequently go dry due to drought and/or water management decisions, and the fate of organic carbon buried upon drying is largely unknown."
Thank you for the feedback, we will revise accordingly.

*P1, L9: You only need to write "mean +/- SE" for the first use. Later, it's implied.
Thank you, we will make this change.

*P2, L1: Are you referring to total burial or burial rates? (I assume the latter.) Reservoirs can have similar OC burial rates as coastal wetlands (which are arguably ocean sediments), so it might be good to mention them. Similar comment for the next sentence about eutrophic reservoirs. What is "high" burial? A value might be useful.
Thank you, we will clarify that we mean total annual burial. We believe that mentioning coastal wetlands is unnecessary since this manuscript is specifically focused on reservoirs, but we will add that the median rate organic carbon burial in eutrophic reservoirs analyzed by Downing et al. 2008 was 2122 g m$^{-2}$ a$^{-1}$.

*P2, L5: Given the ubiquity of reservoirs, "thousands" is insignificant. Consider rephrasing. Also, both the Donchyts and Ragab & Prudhomme references don't seem totally on point; neither reference reservoirs specifically more than a couple of times. They are really about global freshwater trends, which is a different issue. I know there are other references describing annual, if not longer-term drawdowns in reservoirs.
To the best of our knowledge, there is no published estimate for annual or longer-term global water reservoir drawdown or area loss. However, we agree that these two sentences can be improved. We will replace them with the following: *"Meanwhile, reservoirs display highly variable water levels compared to natural lakes (Hayes et al. 2017; Rueda et al. 2006; Zohary and Ostrovsky 2011). Reservoirs often go partially or completely dry due drought and/or water management decisions (Ragab and Prudhomme 2002)."*

*P2, L20: "thought" implies the scientific community thinks this is the case based on evidence. "Commonly assumed" might be a better choice here.
Thank you, we will make this change.

*P2, L25: "shifts in the equilibrium" is unneccesarily vague. Why not just say "precipitation or dissolution"?

Thank you, we will change this to "precipitation or dissolution".

P3, L3: Again, "equilibrium" is a confusing word choice.
Thank you, we will change this to "precipitation or dissolution".

*P3, L17: Was the 10L water filtered? Clarify.
Yes, we will clarify that the water was filtered.

*P3, L23: Missing close parenthesis.
Thank you, we will add close parenthesis.

*P3, L26: How was the water drained? (Drained through the bottom, suctioned off top, etc?)
We will add that the water was suctioned off the top.

*P10, L10: The alkalinity for incubation:wet doesn't seem to be the highest in Table 1.
We will clarify that, among the Incubation cores, Incubation: Wet displayed the highest alkalinity.

*P15, L3: The entire first paragraph belongs in results, not discussion.
Thank you, we will move this paragraph to the results section.

*P15, L13: any more references to include here? One reference doesn't seem sufficient to make the conclusion in the next sentence.
Literature on carbon gas emissions from drying water reservoirs is scarce. To more accurately reflect this, we will rephrase these sentences as follows: *"However, this efflux was lower than that observed in the sediments of another drying reservoir ($900 \pm 150$ mmol $m^{-2}$ $d^{-1}$ in Jin et al. 2016). This contrast suggests the possibility of significant geographic variability in carbon fluxes during reservoir drying, as suggested by Marcé et al. (2019)."*

*P15, L15: This could use more discussion
We will elaborate as follows: *"Similarly, the contrast between the low methane fluxes observed in this study and higher methane fluxes observed in other studies stresses the relevance of local sediment properties in controlling carbon gas fluxes from sediment."*

*P15, L22: The wet-drying core appears no different than the wet incubation core in either Table 1 or Fig 3.
Thank you. We will replace this sentence with *"Both "Incubation: Wet" and "Incubation: Wet-Drying" cores displayed lower organic carbon content than "Initial: Wet" cores (Table 1, Fig. 3)."*

*P16, L5: You might want to reiterate the "previously discussed" evidence here. It's not clear what you're referring to.
Thank you. We reiterate the evidence in our new version of this paragraph, which is provided above in our response to General Comments.

*P16, L6: I was waiting for this to be pointed out. :)
We are glad the reviewer agrees with this observation.

*P16, L8: But there is not large variability within replicates (i.e., errors are low relative to differences between treatments in Table 1). Thus, this explanation for why the dry control has similarly high OC doesn't make sense. I have to disagree that your data don't actually, "supports the presence of an enhanced mineralization process during drying."

OC content in the "Initial: Wet" treatment was $3.52 \pm 0.19$ g/gdw (%), and OC content in the "Initial: Dry" treatment was $3.78 \pm 0.19$ g/gdw (%). Considering that cores within each treatment are from the same location and could thus be considered pseudoreplicates, the standard error of 0.19 g/gdw (%) within both treatments is similar in magnitude to the difference between treatments from different locations, 0.26 g/gdw (%). We will clarify this as follows: *"Considering the relatively large variability among replicate treatment cores collected from the same location (which would therefore be more accurately described as pseudo-replicates), greater spatial replication within the reservoir would likely be necessary to resolve differences in sediment carbon content between the different locations of wet and dry sites."*

*P16, L15: I don't think it's a fair conclusion to say that drying results in an OC loss, because you see the same loss in the wet incubation.

We agree that we cannot conclude that organic carbon loss in "Incubation: Wet-drying" was due to drying because similar loss occurred in the "Incubation: Treatment". We will modify this section as follows to clarify that organic carbon loss occurred in both treatments: *"The observed organic carbon loss from "Incubation: Wet-Drying" and "Incubation: Wet" cores was consistent with the observed carbon gas fluxes in direction, but in both cases organic carbon loss exceeded the amount of carbon gas emitted."*

*P16, L17: These numbers might be better presented in mmol m-2, so as to match the plots and earlier discussion. It took me some mental gymnastics to realize that you're now referring to the actual amount emitted form the 28 cm2 core surface.

Thank you, we will convert to mmol m$^{-2}$.

*P16, L30, L32: Supplementary figure #s missing.

Thank you. We will correct this error, this should refer to Figure S10.

*P17, L1: Did you measure Ca2+? I'm confused.

We will clarify that we would expect elevated $Ca^{2+}$ based on the chemical equation for calcium carbonate dissolution.

*P18, L8: This could be worded better. Yes, bicarbonate is not a long-term sink (geologically, relative to silicate dissolution). But the bicarbonate won't just "transform to CO2", unless via a) carbonate reprecipitation; or b) entering a low pH environment (wherein it still functions as alkalinity).

Thank you. We will rephrase this sentence to: *"While CaCO$_3$ chemical weathering decreases CO$_2$ efflux, it is unlikely to constitute a long-term carbon sink if the groundwater bicarbonate ions produced by dissolution eventually enter the ocean and contribute to carbonate precipitation (Berner et al., 1983)."*

*P18, L10: The link to forest soils is not clear. We're not talking about carbonate dissolution here, are we?

We were not referring to carbonate dissolution, just to carbon sequestration potential. However, in light of this reviewer's comment we believe that this sentence is out-of-context here. We will remove it from our manuscript.

Additional References:

Berner, R.A., A. C. Lasaga, and R. M Garrels. The carbonate–silicate geochemical cycle and its effect on atmospheric carbon-dioxide over the past 100 million years. *Am. J. Sci.* 283 (1983), 641–683.

Cardoso-Silva, S., P. Alves de Lima Ferreira, R. C. Lopes Figueira, D. D. V. Rego da Silva, V. Moschini-Carlos, and M. L. M. Pompêo. "Factors that control the spatial and temporal distributions of phosphorus, nitrogen, and carbon in the sediments of a tropical reservoir." *Environmental Science and Pollution Research* 25, no. 31 (2018): 31776-31789.

Crawley, M. J. *Statistics: An introduction using R*. Wiley, 1988.

Downing, J. A., and L. C. Rath. "Spatial patchiness in the lacustrine sedimentary environment 1." *Limnology and Oceanography* 33, no. 3 (1988): 447-458.

Hayes, N. M., B. R. Deemer, J. R. Corman, N. R. Razavi, and K. E. Strock. "Key differences between lakes and reservoirs modify climate signals: A case for a new conceptual model." *Limnology and Oceanography Letters 2*, no. 2 (2017):47-62.

Mackay, E. B., I. D. Jones, A. M. Folkard, and P. Barker. "Contribution of sediment focussing to heterogeneity of organic carbon and phosphorus burial in small lakes." *Freshwater Biology* 57, no. 2 (2012): 290-304.

Mashhady, A. S., and D. L. Rowell. "Soil alkalinity. I. Equilibria and alkalinity development." *Journal of Soil Science* 29, no. 1 (1978): 65-75.

Moore, G. A. Soilguide (Soil guide): *A handbook for understanding and managing agricultural soils*. Department of Agriculture and Food, Western Australia, Perth. 4343, 2001.

Mudroch, A., and S. D. Macknight. *Handbook of techniques for aquatic sediments sampling.* CRC Press, 1994.

Pittman, B., J. R. Jones, J. J. Millspaugh, R. J. Kremer, and J. A. Downing. "Sediment organic carbon distribution in 4 small northern Missouri impoundments: implications for sampling and carbon sequestration." *Inland Waters* 3, no. 1 (2013): 39-46.

Rueda, F., E. Moreno-Ostos, and J. Armengol. "The residence time of river water in reservoirs." *Ecological Modelling* 191, no. 2 (2006): 260-274.

Suarez, D. L., and J. D. Rhoades. "The apparent solubility of calcium carbonate in soils 1." *Soil Science Society of America Journal 46, no. 4 (1982): 716-722.*

Sumner, M. E. "Measurement of soil pH: problems and solutions." *Communications in Soil Science and Plant Analysis* 25, no. 7-8 (1994): 859-879.

Zohary, T., and I. Ostrovsky. "Ecological impacts of excessive water level fluctuations in stratified freshwater lakes." *Inland Waters 1*, no. 1 (2011): 47-59.